

# The SPARC water vapour assessment II:
# Comparison of annual, semi-annual and quasi-biennial variations in stratospheric and lower mesospheric water vapour observed from satellites

Stefan Lossow[1], Farahnaz Khosrawi[1], Gerald E. Nedoluha[2], Faiza Azam[3], Klaus Bramstedt[3],
John. P. Burrows[3], Bianca M. Dinelli[4], Patrick Eriksson[5], Patrick J. Espy[6], Maya García-Comas[7],
John C. Gille[8], Michael Kiefer[1], Stefan Noël[3], Piera Raspollini[9], Karen H. Rosenlof[10], Alexei Rozanov[3],
Christopher E. Sioris[11], Gabriele P. Stiller[1], Kaley A. Walker[12], and Katja Weigel[3]

[1]Karlsruhe Institute of Technology, Institute for Meteorology and Climate Research, Hermann-von-Helmholtz-Platz 1, 76344 Leopoldshafen, Germany.
[2]Naval Research Laboratory, Remote Sensing Division, 4555 Overlook Avenue Southwest, Washington, DC 20375, USA.
[3]University of Bremen, Institute of Environmental Physics, Otto-Hahn-Allee 1, 28334 Bremen, Germany.
[4]Istituto di Scienze dell'Atmosfera e del Clima del Consiglio Nazionale delle Ricerche (ISAC-CNR), Via Gobetti, 101, 40129 Bologna, Italy.
[5]Chalmers University of Technology, Department of Earth and Space Sciences, Hörsalsvägen 11, 41296 Göteborg, Sweden.
[6]Norwegian University of Science and Technology, Department of Physics, Høgskoleringen 5, 7034 Trondheim, Norway.
[7]Instituto de Astrofísica de Andalucía (IAA-CSIC), Glorieta de la Astronomía, 18008 Granada, Spain.
[8]University of Colorado, Atmospheric Chemistry Observations & Modeling Laboratory, P.O. Box 3000, Boulder, CO 80305-3000, USA.
[9]Istituto di Fisica Applicata del Consiglio Nazionale delle Ricerche (IFAC-CNR), Via Madonna del Piano, 10, 50019 Sesto Fiorentino, Italy.
[10]NOAA Earth System Research Laboratory, Global Monitoring Division, 325 Broadway, Boulder, CO 80305, USA.
[11]York University, Center for Research in Earth and Space Science, 4700 Keele Street, Toronto, Ontario M3J 1P3, Canada.
[12]University of Toronto, Department of Physics, 60 St. George Street, Toronto, Ontario M5S 1A7, Canada.

*Correspondence to:* Stefan Lossow (stefan.lossow@kit.edu)

**Abstract.** In the framework of the second SPARC (Stratosphere-troposphere Processes And their Role in Climate) water assessment (WAVAS-II), the amplitudes and phases of the annual, semi-annual and quasi-biennial variation in stratospheric and

5   lower mesospheric water were compared considering 32 data sets from 13 different satellite instruments. These comparisons aimed to provide a comprehensive overview of the typical uncertainties in the observational database which can be considered in subsequent observational and modelling studies. For the amplitudes, a good agreement of their latitude and altitude distribution was found. Quantitatively there were differences in particular at high latitudes, close to the tropopause and in the lower mesosphere. Here the standard deviation over all data sets typically exceeded 0.2 ppmv for the annual variation and 0.1 ppmv

10   for the semi-annual and quasi-biennial variation. For the phase, larger differences between the data sets were found in the lower mesosphere. Generally the smallest phase uncertainties can be observed in regions where the amplitude of the variability is



large. The standard deviations over all data sets were typically smaller than a month for the annual and semi-annual variation and smaller than 5 months for the quasi-biennial variation. The amplitude and phase differences among the data sets could be explained by a combination of reasons. An important role play temporal variations of systematic errors and differences in the temporal and spatial sampling. In addition, differences in the considered time periods, the vertical resolution of the data,

influences of clouds, aerosols as well as non local thermodynamic equilibrium (NLTE) effects cause differences between the individual data sets.

# 1 Introduction

Water vapour is the most fundamental trace constituent in the troposphere but continues to play an important role in the stratosphere and lower mesosphere. In the lower stratosphere, in particular in the tropics, water vapour is the most important

greenhouse gas strongly affecting global warming at the Earth's surface (Riese et al., 2012). The water vapour feedback, i.e. a warmer climate increases stratospheric water vapour, which in turn leads to an even warmer climate, has been recently estimated to be $0.3 \, \text{W/m}^2$ for a $1 \, \text{K}$ temperature anomaly at $500 \, \text{hPa}$ (Dessler et al., 2013). In addition water vapour is an elemental component of polar stratospheric clouds (PSCs) that are responsible for the strong ozone depletion in the lower stratosphere during winter and spring, especially in the Antarctic. This destruction is based on heterogeneous chemistry that

takes place on the surfaces of the cloud particles. Also, regarding gas phase chemical ozone loss, water vapour plays a decisive role by being the primary source of hydrogen radicals ($HO_x$ = OH, H, $HO_2$) in the middle atmosphere (collectively the stratosphere and mesosphere). Those radicals take part in auto-catalytic cycles that deplete ozone in the stratosphere and mesosphere with a dominant role below ∼25 hPa and above ∼1 hPa (Brasseur and Solomon, 2005; Salawitch et al., 2005). Changes in water vapour would significantly effect both the duration of PSC presence and the production of hydrogen radicals

(Stenke and Grewe, 2005; Khosrawi et al., 2016). In addition to its role in the Earth's radiative budget and middle atmospheric ozone chemistry, water vapour is also an important tracer for transport in the stratosphere and lower mesosphere. Dynamical circulations that can be diagnosed with water vapour include the Brewer-Dobson circulation in the stratosphere and the pole-to-pole circulation in the mesosphere (Brewer, 1949; Remsberg et al., 1984; Mote et al., 1996; Pumphrey and Harwood, 1997; Seele and Hartogh, 1999).

In the stratosphere and lower mesosphere water vapour has two major sources. One is the transport of water vapour into the stratosphere that primarily happens through the cold tropical tropopause layer. Here a large fraction of water vapour is removed due to freeze-drying, leading to a typical entry mixing ratio of about 3.5 ppmv to 4.0 ppmv in the lower stratosphere (Kley et al., 2000; Nassar et al., 2005a). The other major source is the in-situ oxidation of methane within the stratosphere. The importance of this process for the overall water vapour budget increases with altitude and maximises typically in the upper

stratosphere (Le Texier et al., 1988). In the lower mesosphere the oxidation continues but ceases to contribute significantly at pressure levels above about 0.1 hPa due to the low abundance of methane. In addition, water vapour has a minor source in the upper stratosphere due to the oxidation of molecular hydrogen (Sonnemann et al., 2005; Wrotny et al., 2010). The most important sink of water vapour in the stratosphere is the reaction with $O(^1D)$. Towards the upper stratosphere, photodissociation





becomes increasingly important as a sink process and finally dominates in large parts of the mesosphere. An important local and temporary sink, in particular in the Antarctic, is dehydration, the permanent removal of water due to the sedimentation of PSC particles in the polar vortices (Kelly et al., 1989; Fahey et al., 1990). Leaving this aspect aside, the water vapour volume mixing ratio generally increases with altitude in the stratosphere. Typically, an equilibrium between all source and sink processes is

found around the stratopause. The only exception is within the polar vortices where this equilibrium occurs in the middle stratosphere roughly at about 10 hPa. The equilibrium results in a water vapour maximum in the vertical distribution. Here we refer to this as the middle atmospheric water vapour maximum. Above this maximum, water vapour in general decreases with altitude due to the lack of any major source.

Due to the importance of stratospheric and lower mesospheric water vapour, a major research focus over the last two decades

has been to understand long-term changes of this constituent in this altitude region. In the past many observations have indicated an increase in lower stratospheric water vapour (Oltmans and Hofmann, 1995; Oltmans et al., 2000; Rosenlof et al., 2001; Scherer et al., 2008). Hurst et al. (2011) reported an average net increase of ∼1 ppmv in the lower stratosphere based on frostpoint hygrometer observations at Boulder, Colorado (40°N, 105°W) for the time period between 1980 and 2010. More recently Hegglin et al. (2014) and Dessler et al. (2014) analysed different combinations of satellite observations that indicated

negative or no water vapour changes in the lower stratosphere since the late 1980s. The reasons for these observational discrepancies still need to be settled. In the upper stratosphere the satellite observations indicated an increase since the late 1980s in line with the observed increase in methane (Kirschke et al., 2013). In the lower mesosphere fewer results exist. Remsberg (2010) reported an increase in the altitude range between 0.2 hPa to 0.1 hPa considering the time period from 1991 to 2005 and latitudes within 45°S and 45°N. Ground-based microwave radiometer observations at Mauna Loa, Hawaii (20°N, 156°W)

showed an increase in the lower mesosphere throughout the time period from 1996 to 2012 (Nedoluha et al., 2013). On the other hand ,Hallgren et al. (2012) reported a decrease in lower mesospheric water vapour at ALOMAR (Arctic Lidar Observatory for Middle Atmosphere Research, 69°N, 16°E), both in winter and summer, for the time period 1996 to 2010.

For a complete understanding of water vapour changes also short term variability, such as the annual and semi-annual variation or the variation caused by the quasi-biennial oscillation (which we denote here as QBO variation), is an integral part.

The basic aspects of these shorter term variations in stratospheric and lower mesospheric water vapour have been investigated in a number of studies (e.g. Remsberg et al., 1984; Carr et al., 1995; Mote et al., 1996; Nedoluha et al., 1996; Randel et al., 1998; Jackson et al., 1998a, b; Seele and Hartogh, 1999; Schoeberl et al., 2008; Remsberg, 2010; Kawatani et al., 2014). Within the framework of the second SPARC water vapour assessment we have compared characteristics of the annual, semi-annual and QBO variations in the stratosphere and lower mesosphere as derived from a number of satellite data sets. In our comparisons

we focused on the amplitude and the phase of these variability patterns. This work aims to provide estimates of the typical uncertainties in the observational data record that should be taken into account in observational and modelling studies. In the next section we provide a brief overview of the data sets used in our study followed by a description of our analysis approach in Sect. 3. Section 4 focuses on the results for the annual, semi-annual and QBO variation which will be subsequently summarised and discussed in Sect. 5.



## 2  Data sets

Within the second SPARC water vapour assessment a suite of 42 data sets (not including data sets of minor water vapour isotopologues) has been considered, focusing on observations from 2000 until 2014 (Walker and Stiller, in preparation). In the present work we included 32 data sets to compare the characteristics of the annual, semi-annual and QBO variation. A necessary requirement for the analyses was a minimum data set length of one year, ruling out any ILAS-II (effectively 6 months) and SMILES (effectively 4 to 7 months, depending on data set) data sets. Also, we did not include any data sets focusing exclusively on the upper troposphere, which concerns in addition the SMR 501+544 GHz data set (Walker and Stiller, in preparation). Table 1 lists the data sets that have been considered in this analysis. Also given are the corresponding labels used in the figure legends and the effective time periods available for analysis. Overall, data sets from the following 13 instruments have been considered (listed in alphabetical order): ACE-FTS, GOMOS, HALOE, HIRDLS, MAESTRO, MIPAS, MLS (aboard the Aura satellite, not the instrument on the Upper Atmosphere Research Satellite – UARS), POAM III, SAGE II, SAGE III, SCIAMACHY, SMR and SOFIE. For a number of instruments there are multiple data sets based on different retrieval versions, data processors, measurement geometries and spectral signatures to derive the water vapour information. For HALOE, POAM III and SAGE II we considered also the observations before 2000 to have a longer time series for analysis. In the supplement a good first overview of the latitude and altitude coverage of the individual data sets can be found, in particular in Fig. 1. For a complete description of the data sets and their characteristics, the reader is referred to WAVAS-II data set overview paper by Walker and Stiller (in preparation).

## 3  Approach

### 3.1  Data handling

The data sets were initially screened according to the criteria recommended by the data providers, which are summarised by Walker and Stiller (in preparation). The data were then interpolated on to a regular pressure grid using 16 levels per pressure decade, which amounts to a vertical sampling of about 1 km. As top level, a pressure of 0.1 hPa was chosen. Subsequently the data were averaged monthly and for (geographic) latitude bins of 10°. During this step an additional screening was applied to remove unrepresentative observations based on the median and the median absolute deviation (MAD, e.g. Jones et al., 2012). We preferred this method over a screening using the mean and standard deviation as the median is a more robust quantity with regard to extreme outliers. At a given altitude, data points outside the interval [median($\mathbf{x}$) $\pm$ 7.5·MAD($\mathbf{x}$)] were discarded, where $\mathbf{x}$ denotes the data that fall into a particular month and latitude bin. For a normally distributed data set, 7.5·MAD corresponds to about five standard deviations. Hence this is not a very stringent screening. It aims on removing only the most prominent outliers. In the following analyses only averages that were based on at least 20 observations were used, in a further attempt to avoid spurious data. Also averages that were smaller than their corresponding standard error (in absolute terms) were not considered any further.



## 3.2 Comparison parameters

In the comparison of the variability derived from the different data sets, we focused on two parameters, namely the amplitude and the phase (we which collectively denote as variability characteristics). To determine these parameters, multi-linear regressions were employed using the entire time periods of the individual data sets (see last column of Tab. 1). Different regression models were employed to derive the amplitude and phase of the annual, semi-annual and QBO variation. This approach was motivated by the differences in the temporal and spatial coverage of the individual data sets. For the annual variation the regression model consisted only of an offset, a linear term and one sine and one cosine term to parametrise the annual variation:

$$\mathbf{f}(t) = \mathbf{C}_{offset} + \mathbf{C}_{linear} \cdot t +$$
$$\mathbf{C}_{AO_1} \cdot sin(2 \cdot \pi \cdot t/p_{AO}) + \mathbf{C}_{AO_2} \cdot cos(2 \cdot \pi \cdot t/p_{AO}). \tag{1}$$

Here $t$ denotes the time in years, $\mathbf{f}(t)$ the fit of the time series, $\mathbf{C}$ are the regression coefficients of the individual model components and $p_{AO}=1$ year is the period of the annual variation. We followed the method outlined by von Clarmann et al. (2010) to derive those coefficients and used the standard error of the monthly means as statistical weights. Autocorrelation effects and empirical errors were not considered in the regression (Stiller et al., 2012b). Arguably the regression model for the annual variation is quite simplistic. But by that it still allows to derive some reasonable estimates of the annual variation characteristics from the sparser data sets which would not be possible with more advanced regression models. In Sect. 5.3 we discuss the sensitivity of the results by comparing the outcome from different approaches.

The amplitude of the annual variation $\mathbf{A}_{AO}$ can be derived as follows:

$$\mathbf{A}_{AO} = \left| \frac{\mathbf{C}_{AO_2}}{sin[atan(\mathbf{C}_{AO_2}/\mathbf{C}_{AO_1})]} \right| = |\mathbf{A}_{AOsigned}| \qquad \text{for } \mathbf{C}_{AO_1} \neq 0, \mathbf{C}_{AO_2} \neq 0. \tag{2}$$

Be reminded that the amplitude $\mathbf{A}_{AO}$ is only half as large as the variation from maximum to minimum. The phase of the annual variation we represented by the month of a calendar year in which the annual maximum occurred in the regression fit:

$$\mathbf{P}_{AO} = \mathbf{s} \cdot p_{AO} \cdot 12 + 1 - 12 \cdot atan\left(\frac{\mathbf{C}_{AO_2}}{\mathbf{C}_{AO_1}}\right) \cdot \frac{p_{AO}}{2 \cdot \pi}. \tag{3}$$

Here $\mathbf{s}$ is a scaling factor that depends on $\mathbf{A}_{AOsigned}$, with $s=1/4$ if $\mathbf{A}_{AOsigned} > 0$ and $s=3/4$ if $\mathbf{A}_{AOsigned} < 0$. Note, that even though the regression is based on monthly mean data the phase $\mathbf{P}_{AO}$ has a fractional component, which we retain. In this context $\mathbf{P}_{AO}=1.0$ refers to beginning of January, while $\mathbf{P}_{AO}=1.5$ denotes the middle of January.





For the comparison of the semi-annual variation the regression model in Eq. 1 was extended with the corresponding sine and cosine terms ($p_{SAO}$=1/2 year):

$$\mathbf{f}(t) = \mathbf{C}_{offset} + \mathbf{C}_{linear} \cdot t +$$
$$\mathbf{C}_{AO_1} \cdot sin(2 \cdot \pi \cdot t / p_{AO}) + \mathbf{C}_{AO_2} \cdot cos(2 \cdot \pi \cdot t / p_{AO}) +$$
$$\mathbf{C}_{SAO_1} \cdot sin(2 \cdot \pi \cdot t / p_{SAO}) + \mathbf{C}_{SAO_2} \cdot cos(2 \cdot \pi \cdot t / p_{SAO}). \tag{4}$$

Amplitude and phase were derived in the same manner as for the annual variation, with the phase denoting the maximum of the semi-annual fit that is found between January and June.

In the analysis of the QBO variation only those data sets were considered that cover a time period of at least 28 months, which is the average period of this variability pattern (Baldwin et al., 2001). Hence there are no results for the MIPAS data sets that were obtained with full spectral resolution (Fischer et al., 2008), i.e MIPAS-Bologna V5H, MIPAS-ESA V5H, MIPAS-IMKIAA V5H and MIPAS-Oxford V5H. Those four cover only a time period of 21 months (July 2002 to March 2004). In the regression model the QBO variation was described by proxies in form of normalised winds at 50 hPa ($QBO_1$) and 30 hPa ($QBO_2$) over Singapore (1°N, 104°E). The winds at these two pressure levels are approximately orthogonal (e.g. Stiller et al., 2012b) and thus allow the derivation of the QBO amplitude with the same approach as used for the other variability patterns (when normalised). The information on the Singapore winds has been provided by Freie Universität Berlin (http://www.geo.fu-berlin.de/met/ag/strat/produkte/qbo/qbo.dat). For the QBO variation the following regression model was used:

$$\mathbf{f}(t) = \mathbf{C}_{offset} + \mathbf{C}_{linear} \cdot t +$$
$$\mathbf{C}_{AO_1} \cdot sin(2 \cdot \pi \cdot t / p_{AO}) + \mathbf{C}_{AO_2} \cdot cos(2 \cdot \pi \cdot t / p_{AO}) +$$
$$\mathbf{C}_{SAO_1} \cdot sin(2 \cdot \pi \cdot t / p_{SAO}) + \mathbf{C}_{SAO_2} \cdot cos(2 \cdot \pi \cdot t / p_{SAO}) +$$
$$\mathbf{C}_{QBO_1} \cdot QBO_1(t) + \mathbf{C}_{QBO_2} \cdot QBO_2(t). \tag{5}$$

The phase of the QBO variation could be derived in the same manner as done for the annual and semi-annual variation (following Eq. 3) if a certain period $p_{QBO}$ is assumed. However, since the period varies between 22 and 34 months (Baldwin et al., 2001) we decided to express the phase in a different way. For that we calculated the correlation between the QBO fit (i.e. the last row of Eq. 5) and the Singapore winds at 50 hPa shifted by 0 to 30 months, in steps of one month. The shift that maximised the correlation between the two time series was used as the phase.

### 3.3 Standard deviation among data sets

To provide a measure for the typical spread among the data sets the standard deviation for the amplitudes and phases were derived for each variability pattern. We recognise that a Gaussian distribution is not to be expected neither for the amplitudes nor the phases. Still, we assume that the standard deviation could serve as a good proxy of the typical uncertainty in the observational data base. Outliers among the individual data sets can however render this purpose meaningless, in particular for





the amplitudes. Outliers occur especially at altitudes close to the lower and upper boundaries of the individual data sets, where measurement uncertainty is large. To avoid such outliers the amplitudes were screened using once again the median and the median absolute deviation (see Sect. 3.1). For a given latitude and altitude we removed all amplitudes $\mathbf{A}$ that were outside the interval [median($\mathbf{A}$) $\pm$ 7.5·MAD($\mathbf{A}$)]. Overall, between 1.4% and 1.8% of the amplitude data were discarded for the different variability patterns. After the screening the standard deviations for the amplitude were derived. Additionally to the absolute standard deviation $\sigma(\mathbf{A})$, this quantity was also considered in relative terms $\sigma_{rel}(\mathbf{A})$, using the average $\mu(\mathbf{A})$ over all data sets as reference:

$$\sigma_{rel}(\mathbf{A}) = 100 \cdot \frac{\sigma(\mathbf{A})}{\mu(\mathbf{A})}, \tag{6}$$

with

$$\mu(\mathbf{A}) = \frac{1}{\mathbf{n}_s} \cdot \sum_{i=1}^{\mathbf{n}_s} \mathbf{A}_i,$$

$$\sigma(\mathbf{A}) = \sqrt{\frac{1}{\mathbf{n}_s} \cdot \sum_{i=1}^{\mathbf{n}_s} [\mathbf{A}_i - \mu(\mathbf{A})]^2},$$

and $\mathbf{n}_s$ denoting the number of data sets remaining after the amplitude screening.

Different than the amplitude the phase $\mathbf{P}$ has a cyclic nature. Thus applying the same approach as for the amplitudes would occasionally result in misleading estimates of the standard deviation. Therefore a different approach was chosen which is based on the phase difference to a chosen reference data set $\mathbf{P}_{ref}$. This difference $\Delta\mathbf{P} = \mathbf{P} - \mathbf{P}_{ref}$ has always been adapted so that it did not exceed the interval $\pm 6/3/15$ months for the annual/semi-annual/QBO variation (by adding $\pm 12/6/30$ months in cases it was necessary). Different to the amplitude we only focused on the absolute standard deviation (in months):

$$\sigma(\Delta\mathbf{P}) = \sqrt{\frac{1}{\mathbf{n}_s - 1} \cdot \sum_{i=1}^{\mathbf{n}_s - 1} [\Delta\mathbf{P}_i - \mu(\Delta\mathbf{P})]^2}, \tag{7}$$

with

$$\mu(\Delta\mathbf{P}) = \frac{1}{\mathbf{n}_s - 1} \cdot \sum_{i=1}^{\mathbf{n}_s - 1} \Delta\mathbf{P}_i.$$

The phases corresponding to the amplitudes screened were discarded as well. Thus $\mathbf{n}_s$ is used again to describe the effective number of data sets, but is diminished by 1 to account for the reference data set. Given the approach, the standard deviation $\sigma(\Delta\mathbf{P})$ is dependent on the reference data set. Tests with different reference data sets showed a consistent picture of the



main features. The largest quantitative uncertainties were found in the lower mesosphere towards the uppermost altitude levels considered here. As reference data sets were primarily those reviewed that have a more or less complete coverage of the latitude-altitude domain considered here, i.e. the MIPAS V5H NOM or V5R NOM data sets from the different processors and the two MLS data sets. To this end, the MLS v4.2 data set was chosen as reference since it likely better represents the lower mesosphere compared to the MIPAS data sets which are influenced by NLTE effects in this altitude region (see also discussion in Sect. 5.2).

## 4 Results

In this section, the comparison results for the annual, semi-annual and QBO variation will be described. For every pattern we start with an example of the typical latitude-altitude distribution for the amplitude and phase to describe briefly the most prominent features. In the supplement, such plots for all data sets considered in this work are provided. Thereafter the variability characteristics are shown at specific latitudes and altitudes, combining all data sets, allowing a direct comparison and the estimation of typical uncertainties. Finally a summary, in the form of standard deviations for the amplitude and phase based on all data sets, as described in Sect. 3.3, are provided.

The main focus will be on the stratosphere, with some discussion of the lower mesosphere. Upper tropospheric results are visible in many figures, but will not be discussed. The mean tropopause for the time period 2000 – 2014 based on MERRA (Modern Era Retrospective-Analysis for Research and Applications, Rienecker et al., 2011) reanalysis data is indicated in the corresponding figures for guidance.

### 4.1 Annual variation

### 4.1.1 General characteristics

Figure 1 shows an example of the amplitude (left panel) and phase (right panel) of the annual variation in water vapour as function of latitude and altitude, revealing several key features of this variation in the stratosphere and lower mesosphere. These features are the result of periodicity in a combination of processes: vertical transport, dehydration at the tropical tropopause, dehydration in the polar regions and methane chemistry. We have highlighted five features using black and red boxes (the colour variation is for better contrast) which are briefly described below:

1. This feature, the "atmospheric tape recorder" (Mote et al., 1996) is a consequence of the annual variation of dehydration at the tropical tropopause due to the annual variation of tropical tropopause temperatures. The signal is transported upwards by the ascending branch of the Brewer-Dobson circulation and maintains its integrity because of the subtropical mixing barriers in the lower stratosphere. In terms of the amplitude this feature is less remarkable. However, the phase exhibits the characteristic upward progression as the tape recorder signal moves from the tropopause up to a pressure level slightly below 10 hPa typically within 12 to 18 months.





2. The amplitude of the annual variation exhibits a distinct local maximum in the southern tropical and sub-tropical middle and lower stratosphere. The phase plot indicates that this variation has its annual maximum in late summer and early autumn. In the northern hemisphere a weaker counterpart can be found (not marked). The annual cycle of vertical transport within the Brewer-Dobson circulation essentially explains the annual cycle in water vapour. However to account for the full amplitude of this annual variation in water vapour also contributions from chemistry and eddy transport are necessary, while meridional advection acts to reduce the amplitude. The inter-hemispheric differences are primarily due to differences in the downwelling of moister air from above which is characterised by strong water vapour gradients due to the methane chemistry. A full description of this feature will be given by Lossow et al. (in preparation).

3. This amplitude maximum is due to the annual cycle of dehydration caused by PSC ice particles (Kelly et al., 1989; Fahey et al., 1990). These particles form in the lower stratosphere during polar winter and early spring. Once these grow to substantial sizes they sediment out resulting in a substantial reduction of water vapour in the ambient atmosphere. As these clouds are more common in the Antarctic than in the Arctic, this feature is more pronounced in the southern than in the northern hemisphere (not marked). As the Antarctic dehydration occurs on a rather constant level for a few months this feature is not a classic (sinusoidal) annual variation but more of seasonal characteristic.

4. This high amplitude in the annual variation here is caused by the annual cycle in the vertical transport at high latitudes. During winter, the middle atmospheric water vapour maximum is shifted down to about 10 hPa (Nassar et al., 2005b; Lossow et al., 2009) while during summer it rises up to the stratopause (Seele and Hartogh, 1999). Correspondingly the annual maximum is found during winter. The amplitude of the annual variation is larger in the Antarctic than its northern counterpart due to the stronger and more stable polar vortex in the southern hemisphere.

5. A large amplitude in the annual variation can be found in the lower polar mesosphere in both hemispheres. The reasons for this feature are very similar to those for key feature #4, except that it occurs at altitudes above the middle atmospheric water vapour maximum where its abundance typically decreases with altitude. During winter dry air from the upper mesosphere descends within the polar vortex, while during summer and early autumn moist air from below is transported upwards. In addition, in summer more water vapour is produced from methane oxidation due to higher insolation (Bates and Nicolet, 1950; Le Texier et al., 1988).

As visible in the supplement (see Fig. 1) these features are well depicted in most data sets, even though quantitative differences exist. Depending upon adequate observational coverage all features can be found (like in the MIPAS and MLS data sets) or a subset of those.

### 4.1.2 Examples at specific latitudes and altitudes

Here we focus on the annual variation characteristics around the equator (key feature #1), in the Antarctic (key features #3 – 5) and at a pressure level of 1 hPa (key feature #5, slightly above key feature #2). In Fig. 2 the characteristics are shown for the latitude range between 5°S and 5°N as function of altitude. The dark grey dashed line indicates the climatological tropopause





altitude from the MERRA reanalysis. Overall 24 data sets have been included in this figure. The POAM III, SCIAMACHY (using occultation), SAGE III and SOFIE data sets have no tropical observations while the GOMOS and MAESTRO data sets do not have sufficient temporal coverage to estimate the annual cycle. Overall, there is a good consistency in the vertical structure of the amplitude, showing clearly one maximum close to the tropopause and another one around 3 hPa (related to

key feature #2). Close to the tropopause, the amplitudes exhibit a large spread (∼0.6 ppmv) among the data sets, while in the phase such deviations are not that obvious. The phase progression within the tape recorder can be seen very well, only the SCIAMACHY limb, SMR and HIRDLS data sets exhibit distinct differences. They can all be attributed to the increased uncertainty in the data sets close to their vertical retrieval boundaries. Also, the phase progression is quite steep and deviations of one month in the occurrence of the annual maximum are not uncommon. At 10 hPa the spread among the data sets is quite

small in terms of the amplitude (less than 0.1 ppmv) while larger deviations exist in the phase as there is a transition from the tape recorder to a different variability pattern. Up to 1 hPa solely the ACE-FTS data sets exhibit some more obvious deviations. For the amplitude it is only the ACE-FTS v2.2 data set, in terms of the phase both data sets are affected. In the lower mesosphere the data sets exhibit clearly less consistent results. This concerns in particular the phase. There is certainly some clustering of data sets, but close to 0.1 hPa there is a tendency to show the annual maximum either in the middle of the year or around the

turn of the year. A similar behaviour can also be seen in the amplitude of the annual variation, with some data sets indicating a small annual variation (up to 0.1 ppmv) and other data sets showing a more pronounced variation (around 0.3 ppmv). Beyond that, the MIPAS-Oxford data sets exhibit very large amplitudes in the order of 0.8 ppmv to 1 ppmv.

Figure 3 shows the characteristics of annual variation in the latitude range between 85°S and 75°S, again considering 24 data sets. Here the HIRDLS, SAGE III and SCIAMACHY solar occultation data sets have no observations, while the

GOMOS, HALOE and SAGE II data sets had no sufficient temporal coverage for the analysis. Compared to the results for the tropics shown in the last figure, overall a larger spread among the data sets is visible. This can be most prominently observed in the amplitudes below about 20 hPa (related to key feature #3) where the spread is more than 1 ppmv. For the phase, the consistency in this altitude range is better. Many data sets show the annual maximum in the early months of the year, yet differences of up to 3 months are visible. Around 20 hPa, a rather abrupt phase transition can be found towards the annual

maximum occurring during the middle of the year, i.e. austral winter. This is accompanied by a large spread as this transition height varies for the individual data sets. Above 20 hPa, a good agreement among the data sets can be observed in the vertical structure of amplitude and phase (key feature #4 and #5). Still, the spread is substantial. Up to about 0.5 hPa, the spread is in the order of 0.4 ppmv for the amplitude and 2 to 3 months for the phase. Higher up the spread increases even more, mainly due to large deviations by various MIPAS data sets. Between 20 hPa and 0.3 hPa the SCIAMACHY lunar, MIPAS-IMKIAA V5H

and MIPAS-Oxford V5H data sets deviate obviously from the other data sets.

Figure 4 compares the data sets at an altitude of 1 hPa, indicating that the majority of those capture well the latitude dependence of the annual variation in terms of amplitude and phase. Results from 27 data sets are shown, not including the GOMOS, HIRDLS, MAESTRO, SCIAMACHY limb and SMR 544 GHz data sets as they do not cover this high altitude. The largest spreads in amplitude (0.5 ppmv to 0.7 ppmv) occur at the highest latitudes where also the annual variation maximises (key fea-

ture #5). In particular the MIPAS-IMKIAA V5H and V5R NOM, POAM III and SCIAMACHY solar Onion data sets exhibit



high amplitudes which cause the large spread. Without those data sets the spread would be typically in the order of 0.3 ppmv to 0.4 ppmv. The annual maximum in the polar regions can be found in late summer and early autumn at this altitude and a typical spread of 2 months among the data sets is visible. The amplitude and the phase exhibit a distinct structure in their latitude distribution at mid and low latitudes. Amplitude minima at 40°S/N and the equator as well as amplitude maxima around the

subtropics (in the southern hemisphere related to key feature #2 that peaks at ∼3 hPa) can be observed quite consistently. The spread among the data sets varies between 0.1 ppmv and 0.2 ppmv depending on latitude. Larger deviations can be found for both ACE-FTS data sets. The SMR 489 GHz data set shows generally higher amplitudes in the southern tropics while there is a similar tendency for the MIPAS-Oxford V5H data set in the subtropics of that hemisphere. Around 20°N, the HALOE v20, SAGE II and SMR 489 GHz data sets deviate from the general latitudinal behaviour indicating a minimum which also coincides

with obvious deviations in the phase. These data sets show the annual maximum in May and June while majority of data sets do so around the turn of the year. In general, the phase passes through the entire year from 45°S to 45°N with water vapour maxima around the turn of the year in the subtropics and in the middle of the year in the inner tropics. The typical spread is in the order of 1 to 2 months in this latitude range.

### 4.1.3   Standard deviation among data sets

Figure 5 quantifies the spread among the different data sets in form of the standard deviation for the amplitude and phase of the annual variation. The upper panel shows the standard deviations of the amplitude in absolute terms (see Sect. 3.3). Here the largest standard deviations are typically found around the tropopause and the polar regions, indicating the least agreement between the data sets. In particular, this concerns the lower and middle stratosphere in the Antarctic and the lower mesosphere in both hemispheres. Here often the standard deviations exceed values of 0.2 ppmv. In the other regions standard deviations

in the order of 0.1 ppmv are common. The middle panel focuses on the standard deviations of the amplitude in relative terms, normalised by the mean amplitude from all data sets. In this domain a somewhat different picture reveals. The areas with large standard deviations in absolute terms often do not coincide with the areas showing the largest relative standard deviations. Those lie typically in the regions where the amplitude of the annual variation is small, i.e. predominantly in the mid and low latitudes. Here relative standard deviations of 50% and more can be found occasionally. In the polar upper stratosphere and

lower mesosphere the relative standard deviation is often smaller than 40%, in particular in the Arctic. In the area related to key feature #2 in the southern upper tropical and subtropical stratosphere, the relative deviations are within 30%.

The lower panel of Fig. 5 shows the standard deviation of the phase difference with respect to the MLS v4.2 data set. The overall pattern is very similar to the relative standard deviation in the amplitude. At polar latitudes above 10 hPa (key feature #4 and #5) as well as in tropical and subtropical upper stratosphere in the southern hemisphere (key feature #2), standard deviations

of less than 1 month can be observed. Standard deviations larger than 2.5 months are visible most prominently in the lower stratosphere in the Antarctic, at latitudes polewards of 45°N between 50 hPa and 10 hPa and moreover in the tropical lower mesosphere.



## 4.2 Semi-annual variation

### 4.2.1 General characteristics

Figure 6 shows an example of the characteristics of the semi-annual variation in water vapour in the same manner as done for the annual variation in Fig. 1. Note, that the amplitude range is smaller than that for the annual variation. As before key features are marked by boxes and are briefly described below:

1. In the tropical upper stratosphere the semi-annual variation is the most important mode of annual variability. The larger amplitudes in water vapour are due to vertical and meridional advection induced by the stratospheric semi-annual oscillation in the zonal wind (SSAO). This oscillation is forced by a combination of processes. The easterlies accelerations are due to the meridional advection of easterlies from the summer hemisphere across the equator as well as the eddy momentum deposition from breaking planetary waves which have been ducted in the tropics. Momentum deposition from the interaction of vertically propagating ultra-fast Kelvin waves and internal gravity waves with the mean flow cause the westerly accelerations (Hamilton, 1998). The annual maxima in water vapour are typically found before the solstices (Randel et al., 1998; Jackson et al., 1998a).

2. This maximum is related to the dehydration due to PSCs (see key feature #3 for the annual variation) and the annual cycle in the vertical transport. At the beginning of the year, the water vapour volume mixing ratios show a minimum in these altitudes. Towards wintertime water vapour increases due to the downwelling of moist air. Soon after temperatures allow the wide-spread formation of PSCs, dehydration sets in causing a deep minimum in water vapour during austral winter and spring. After September, the water vapour volume mixing ratios increase again as the PSC influence becomes less. Typically in November a small maximum can be observed before the upwelling in summer decreases water vapour again. Thus, this feature is not a semi-annual variation in the classical sense as also noted for the corresponding key feature #3 of the annual variation. In the sinusoidal regression approach employed here this semi-annual variation also acts as a correction of the non-sinusoidal shape of the annual variation.

3. Due to the annual cycle in the vertical transport in the polar regions, the altitude of the middle atmospheric maximum in the vertical water vapour distribution shifts from roughly 10 hPa in winter up to the stratopause in summer (see key feature #4 for the annual variation). This shift gives rise to a semi-annual variation at altitudes in-between. Maxima in the annual variation occur in spring and autumn in the transition of the vertical winds from winter to summer conditions and vice versa.

4. Increasing amplitudes can be observed towards the middle mesosphere in the polar regions. A strong maximum can be observed in summer (key feature #5 of the annual variation) due to the upwelling of moist air but there is also a small maximum or plateau in late winter and early spring. In the Arctic, a large contribution to this behaviour arises from sudden stratospheric warmings that have been relatively frequent since the new millennium (e.g. Manney et al., 2008; Orsolini et al., 2010; Straub et al., 2012). These warmings lead to a breakup of the vortex breakup for a short period of





time, during which moister air is advected from lower latitudes and altitudes. In the Antarctic this behaviour is due to a special condition in the wave forcing of the mesospheric pole-to-pole circulation, which is a topic of current research.

5. Above key feature #1, the amplitude of the semi-annual variation decreases significantly. Higher up in the middle mesosphere the amplitudes increase again. The annual maxima typically occur around equinox, thus phase-shifted relative to the prominent semi-annual variation in the upper stratosphere (Jackson et al., 1998a; Lossow et al., 2008). This feature is due to the meridional and vertical advection caused by the mesospheric semi-annual oscillation in the zonal wind (MSAO). The forcing of this oscillation is attributed to the momentum deposition from a broad spectrum of vertically propagating gravity and high-speed Kelvin waves excited in the lower atmosphere. The filtering of these waves by the SSAO allows only waves to propagate further up that have opposite horizontal propagation directions than the zonal wind direction in the upper stratosphere. Thus the wave filtering accounts for the phase shift between the SSAO and MSAO (Hamilton, 1998). In the example the amplitude of the semi-annual variation in water vapour is relatively small but other data sets exhibit a significantly larger variation, as visible in the left panel of Fig. 7.

The semi-annual features highlighted in Fig. 6 are evident in most of the data sets (see supplement, Fig. 2), as noted for the annual variation. Some of the quantitative differences are discussed in the following subsection.

### 4.2.2 Examples at specific latitudes and altitudes

For the semi-annual variation, we detail characteristics in the tropics (key features #1 and #5), the Arctic (key features #3 and #4) and at a pressure level of 2.4 hPa where the semi-annual variation is strong in the tropics (key feature #1). Figure 7 shows the amplitude and phase for the semi-annual variation in the inner tropics. Compared to the corresponding figure for the annual variation (Fig. 2), both ACE-FTS data sets are missing. They provide observations only during four months a year in this latitude range which is not sufficient to derive meaningful results for the semi-annual variation. As for the annual variation, the altitude dependence of the amplitude is fairly consistent among the various data sets. The two key features in this latitude range are well depicted by all data sets but a large spread among them can be found. For key feature #1, in the upper stratosphere, the amplitudes range from 0.2 ppmv to 0.4 ppmv, with several of the MIPAS V5H data sets at the upper end of this interval. For key feature #5, in the lower mesosphere, the spread is even larger, in particular towards the upper boundary of the figure. The largest amplitudes (up to 0.6 ppmv) are again indicated by several MIPAS data sets, while for example the HALOE, MLS and SMR 489 GHz data sets show small amplitudes (up to 0.15 ppmv). Around 30 hPa the amplitude drops close to zero which is reflected by most data sets with a low degree of uncertainty (about 0.05 ppmv spread). The only exceptions are HIRDLS and SAGE II which exhibit amplitudes between 0.1 ppmv and 0.2 ppmv as well as the SMR 544 GHz data set which shows even higher amplitudes. The phase of the semi-annual variation shows a distinct progression with altitude in the lower stratosphere, similar to that characteristic for the annual variation relating to the atmospheric tape recorder. The spread among the data sets can be as large as about 1.5 months. Better agreement can be observed in the altitude range between 15 hPa and 1 hPa where the spread is typically less than a month. In the lower mesosphere a large uncertainty in the phase can be found in conjunction with the large spread in amplitude.





Fig. 8 shows the characteristics for the semi-annual variation in the latitude range between 70°N and 80°N, considering 24 data sets. The SCIAMACHY occultation data sets have no observations in this latitude range. In addition, the ACE-FTS, MAESTRO, GOMOS and POAM III data sets are not included. These data sets provide some data in this latitude range, but not an adequate number of months to derive the characteristics properly. In this latitude range large spreads can be observed, both in amplitude and phase. The spread of the amplitudes typically amounts to several tenths of a ppmv at all altitudes. This makes it difficult to distinguish for example key feature #3 in the upper stratosphere, while key feature #4 in the lower mesosphere can be recognised more easily. Likewise it is difficult to discern the altitude dependence of the phase. In the right panel of Fig. 6 this is more easily seen.

The latitude dependence of the semi-annual variation characteristics at an altitude of 2.4 hPa (key features #1 and #3) is shown in Figure 9. In total, 26 data sets are examined. As for Fig. 4 the GOMOS, HIRDLS, MAESTRO, SCIAMACHY limb and SMR 544 GHz data sets do not cover this altitude, while the SCIAMACHY lunar data set was not sufficient for analysis here. The comparison exhibits a large degree of consistency in the latitude distribution of amplitude and phase among the different data sets. The spread in the amplitudes is typically in the order of 0.1 ppmv. Only at high latitudes and the equator, is this value exceeded. At the equator, the spread amounts to 0.2 ppmv and is even larger in the Arctic and Antarctic. This is mainly due to several data sets that show rather large (all MIPAS-Oxford, POAM III, SAGE II and SAGE III data sets) or rather small (MIPAS-ESA V5H, MIPAS-Oxford V5H, SAGE III and SMR 489 GHz data sets) amplitudes. For the latter group this often coincides with a minimum at high altitudes while the other data sets show a clear maximum. At mid-latitudes the ACE-FTS data sets exhibit occasionally quite large amplitudes. The spread in the phase is relatively small, typically a little bit less than a month. Only in the Antarctic and polewards of 45°N, a larger spread can be found.

### 4.2.3    Standard deviation among data sets

Based on the combination of results from all data sets, Fig. 10 shows the standard deviations of the variability characteristics for the semi-annual variation. Many aspects of the distribution are similar to the results obtained for the annual variation shown in Fig. 5. The lowest standard deviations in absolute terms for the amplitude (upper panel) can be observed in the stratosphere and lowermost mesosphere at mid and low latitudes. Here the standard deviations are typically less than 0.06 ppmv. In the Antarctic between about 60 hPa and 20 hPa (key feature #2) very large deviations in the amplitudes can be found, often exceeding 0.2 ppmv. A similar picture can be seen close to the upper altitude boundary considered here. Values in excess of 0.2 ppmv are observed in the polar regions and the tropics. In the latter region also the relative standard deviations of the amplitude (middle panel) are large, i.e. typically above 60%. Overall, in relative terms, the lowest standard deviations, and thus the best agreement between the data sets, are observed in relation to key feature #1 in the upper tropical and subtropical stratosphere. Here values of up to 30% can be seen. At lower altitudes, the relative standard deviation of the amplitude is typically between 50% and 100%, despite the small absolute standard deviations. The opposite relation can be found for the polar lower stratosphere in the southern hemisphere (key feature #2). Towards the Arctic, in the lower stratosphere larger standard deviations (>60%) are visible, while in the polar upper stratosphere and lower mesosphere they are typically between 30% and 60%. On average, the absolute standard deviations for the amplitude are smaller for the semi-annual variation than for the annual variation. For





the relative standard deviation, the situation is the opposite. For the phase, the satellite data sets agree best in the tropical and subtropical upper stratosphere (key feature #1), where the standard deviations are typically less than a half month. The largest standard deviations, in the order of 2 months, can be observed above in the lower mesosphere.

## 4.3 Quasi-biennial variation

### 4.3.1 General characteristics

Various characteristics of the QBO variation in water vapour are shown in Fig. 11. The phase is given as the shift of the QBO regression fit for which the correlation to the Singapore winds at 50 hPa maximised. The amplitude of the QBO variation exhibits three key features which are described briefly below:

1. In the tropical lower and middle stratosphere, the QBO variation exhibits a slightly enhanced amplitude. This region comprises the area where the signal of the QBO is strongest in the zonal wind. QBO variations in the vertical wind affecting the transport of water vapour as well as the advection of QBO induced anomalies in the stratospheric entry water vapour contribute to this feature (Kawatani et al., 2014). The phase shift with respect to the Singapore winds at 50 hPa is altitude-dependent and changes from about 12 months at the bottom of this range to more than 20 months at the top.

2. A much larger amplitude is found in the tropical upper stratosphere. This variation is, to a large extent, due to transport of water vapour by the QBO-induced anomalies of the residual circulation (e.g. Geller et al., 2002). A small phase shift in the order of a few months can be found. This feature has a profound effect on key feature #1 of the semi-annual variation in the upper tropical stratosphere (Jackson et al., 1998a).

3. An enhanced amplitude is found in the polar regions. In the Antarctic the maximum can be observed typically close to 10 hPa. In the Arctic the variation is smaller and the maximum has a tendency to occur higher up than in the southern hemisphere. This feature reflects the influence of the QBO on the polar regions, likely induced through QBO variations in the downwelling conditions during winter and the upwelling conditions during summer. The variation is more pronounced in the Antarctic due to stronger variations in vertical velocity in this hemisphere. As for key feature #1 the phase shift has a pronounced altitude-dependence.

### 4.3.2 Examples at specific latitudes and altitudes

For the QBO, we focus again on the tropics (key features #1 and #2) and show an example at an altitude of 7.5 hPa which addresses key feature #3. The characteristics in the tropics are shown in Fig. 12 considering 18 data sets. Compared to Fig. 7, which shows the corresponding characteristics for the semi-annual variation, all MIPAS V5H data sets are missing because none covers a complete QBO cycle (see Tab. 1 and Sect. 3.2). In Fig. 12, the two tropical key features (#1 and #2) are clearly visible in the amplitude and phase. Around the lower amplitude peak, the HIRDLS, SAGE II and SMR 544 GHz data sets indicate much larger amplitudes than the other data sets and thus increase the spread considerably. The spread of the phase shift





on the other hand is small, amounting only to a few months. This is also a characteristic value for the key feature #2 in the upper stratosphere. Most data sets indicate here a QBO amplitude on the order of 0.5 ppmv. Both MLS data sets exhibit somewhat lower amplitudes, while the SMR 489 GHz data set shows a relatively weak QBO variation peaking at only 0.25 ppmv. Up to a level of 0.3 hPa there is a rather good agreement in the amplitude of the QBO variation. Higher up, several data sets indicate a

quite substantial increase of the amplitude with altitude. This concerns the MIPAS-Bologna and MIPAS-ESA data sets as well as the MIPAS-Oxford V5R MA data set. The MIPAS-Oxford V5R NOM data set shows such increase only up to ∼0.15 hPa. Thus the spread at 0.1 hPa amounts to about 0.3 ppmv. Focusing on the phase, the same kind of clustering as in the amplitude cannot be found. The majority of data sets indicate a phase shift between 6 and 11 months in the lower mesopause with little altitude-dependence in general. Yet, the spread is large as the shift ranges from 0 up to 24 months. The SMR 489 GHz alternates

between these extreme values depending on altitude. The MIPAS-Bologna datasets consistently exhibit phase shifts between 15 months and 20 months, while the MIPAS-Oxford V5R MA data set often shows no shift at all.

A final example is given in Fig. 13 that considers the latitude-dependence of the QBO variation at 7.5 hPa. This level cuts through key feature #3 in the Antarctic and is very close to the lower boundary of key feature #2 in the tropics. Overall 24 data sets are compared, including the GOMOS data set. The previous latitude cross sections (Fig. 4 and 9) focused on altitudes

above the upper retrieval boundary of the GOMOS data set. The amplitude exhibits a spread of about 0.3 ppmv in the Antarctic. The different MIPAS data sets can be found consistently on the upper end while the remaining data sets tend to exhibit small amplitudes. In particular, ACE-FTS v2.2 and SAGE II show very low amplitudes between 75°S and 60°S. Towards the southern mid-latitudes and sub-tropics, the spread decreases and for most data sets the amplitudes agree within 0.1 ppmv. Here, both HALOE data sets and SAGE II indicate often the lowest amplitudes. Within the tropics again large variability can be found

and the spread is similar to that in the Antarctic. As before, a number of MIPAS data sets account for the highest amplitudes which are likely related to key feature #2. Around 30°N the data sets indicate consistently a maximum in the amplitude. The spread is on the order of 0.1 ppmv excluding the very high amplitudes derived from the GOMOS and SAGE II data sets. The northern mid and high latitudes are characterised by good agreement between the data sets, typically within 0.1 ppmv. As in the southern hemisphere the HALOE and SAGE II data sets exhibit the lowest amplitudes in the mid-latitudes while in the Arctic

SAGE III shows significantly larger amplitudes than the other data sets. While there is less spread in the QBO amplitude in the Arctic than Antarctic, the situation is opposite for the phase shift. In the Antarctic the phase shift for all data sets is typically within a few months while in the Arctic it exceeds 5 months. In both polar regions there are few data sets that occasionally indicate no phase shift resulting in very large deviations from the other data sets. In the southern hemisphere, this applies to the SAGE II data set and in the northern hemisphere this involves the SCIAMACHY solar occultation data sets. In the northern

mid-latitudes and subtropics, the spread is similar as in the Arctic and again a few data sets indicate no phase shift and thus a large deviation. In the southern mid-latitudes and subtropics the situation is more complicated. There is one group of data sets that consistently show phase shifts between 20 months and 25 months while the other group of data sets exhibits phase shifts of less than 10 months. Also in the tropics there is a significant spread with phase shifts ranging from 0 to 15 months. For the latitude range between 5°S and 10°N, most data sets however indicate a phase shift of 6 months.



### 4.3.3   Standard deviation among data sets

As for the other variability patterns Fig. 14 summarises the standard deviation among the data sets in terms of the QBO variation. The absolute standard deviations of the amplitude (upper panel) reflect almost the QBO amplitude itself i.e. larger standard deviations are found in regions where the QBO variation exhibits larger amplitudes. Typically the standard deviations

amount up to 0.12 ppmv. There is also some coherence between the QBO amplitude and the relative standard deviations (middle panel). At many latitudes and altitudes, the relative standard deviations of the amplitude are around 50%. Lower values can be observed in the area related to key feature #2 in the upper tropical and subtropical stratosphere as well as in larger parts of the middle stratosphere (partly related to key feature #3). Here the relative standard deviations are typically less than 30%. In comparison with the annual and semi-annual variation the absolute standard deviations of the QBO amplitude are smaller on

average. In terms of the relative standard deviations the QBO variation ranks between the annual and the semi-annual variation. As for the other variability patterns, the standard deviation of the QBO phase (lower panel) largely correlates with the relative standard deviation of the amplitude. Low standard deviations can be observed in the middle and upper tropical stratosphere, with values of less than 3 months. In the lower mesosphere, especially in the northern hemisphere, the standard deviations maximise at about 5 to 10 months, indicating the lowest agreement between the data sets.

## 5   Discussion

The amplitudes and phases of the annual, semi-annual and QBO variation in stratospheric and lower mesospheric water vapour from 32 data sets were compared to provide estimates of the typical uncertainties in these essential quantities in the observational data record. In this section, the results are first summarised. Thereafter possible reasons for differences among the data sets are discussed and the sensitivity of results is addressed, by considering other approaches to derive the variability

characteristics.

### 5.1   Summary of results

Overall, good agreement in the latitude-altitude distributions of the variability characteristics was found, in particular for the amplitudes for all modes of variability considered. The key features for the individual variability patterns briefly described in Sects. 4.1.1, 4.2.1 and  4.3.1 were observed in most data sets. However, occasionally obvious quantitative differences were

visible between the data sets as illustrated by various examples in the previous section.

To summarise the results for the amplitudes of the different variability patterns standard deviations among all data sets were derived, in absolute terms but also relative to the average over all data sets (Figs. 5, 10 and 14). Depending upon which type of standard deviation is considered, the conclusions are different. A common pattern for the absolute standard deviations was to observe low deviations where also the variability is low. In contrast, around the key features larger standard deviations in

absolute terms could be found. Data set differences were most prominent in the tropopause region, high latitudes and altitudes close to the upper limit of the analysis (0.1 hPa). Overall, for the annual variation, the occurrence rate of absolute standard devi-




ations smaller than 0.1 ppmv was almost 50% considering all latitudes and altitudes above the tropopause. The corresponding values for the smaller semi-annual and QBO variations were 73% and 86%, considering only a standard deviation of 0.05 ppmv yielded occurrence rates of 42% and 37%, respectively. In the vicinity of the key features the relative standard deviation was often smallest. Large values, on the other hand, could often be observed in areas where the variability is low. Here the most

problematic regions were generally the lower stratosphere at mid and high latitudes as well as the lower mesosphere in the tropics and sub-tropics. Above the tropopause 68% of the standard deviations were smaller than 50% for the annual variation. For the semi-annual variation, 51% of the standard deviations were below this threshold and for the QBO variation it was 61%.

The phases of the different variability patterns exhibited the smallest uncertainties around the key features and larger uncertainties elsewhere. This strongly resembles the pattern found for the relative standard deviation of the amplitudes. For a

summary, we derived here the standard deviations over the phase difference to a reference data set, which was chosen to be the MLS v4.2 data set. Taking again into account all latitudes and altitudes above the tropopause the occurrence rate of phase differences within ±1 month was slightly more than 60% for the annual variation. For the semi-annual variation 47% of the absolute standard deviations were smaller than half a month and for the QBO variation 72% were within ±5 months. Turning to the standard deviation itself, 39% of its values were below 1 month for the annual variation. For the semi-annual variation,

the percentage was 16% for half a month and, for the QBO variation, 66% of the standard deviations were below 5 months.

## 5.2 Reasons for differences

There are a number of reasons that potentially explain the differences observed among the data sets. Most prominently (1) different measurement time periods, (2) different sampling in time and space, (3) temporal variations of systematic errors, (4) differences in vertical resolution, (5) influences of clouds and aerosols and (6) NLTE effects, which are considered in the

following subsections. To quantify the impact of some of the aspects listed above we performed sensitivity studies (not shown). Changes are always given as sensitivity test results minus the results presented in Sect. 4. The latter results we also used as the reference for relative changes.

### 5.2.1 Different measurement time periods

In our analysis we always used the entire time periods covered by the individual data sets. Differences in those, combined

with inter-annual variability and changes in the QBO, impact the results. This aspect was investigated by comparing results where the entire measurement periods of the data sets were used and results where only observations between January 2006 and December 2011 were considered. In total 19 data sets (ACE-FTS, MAESTRO, MIPAS V5R NOM, MIPAS V5R MA, MLS, SCIAMACHY and SMR) contributed to this sensitivity study. The expected improvement of the consistency among the data sets was only found for the QBO variation. In particular, the spread in the QBO phase was reduced. For the amplitudes,

the improvement was less considerable. Given the role of the QBO for the inter-annual variability, this result is not surprising. Focusing on a consistent time period will also eliminate the transient changes in QBO strength and period (Baldwin et al., 2001; Kawatani and Hamilton, 2013). Based on the absolute standard deviation, the spread among the data sets increased slightly overall for the annual variation, both in terms of amplitude and phase. Quite prominently an improvement was found



in the tropical middle and upper stratosphere, while other regions predominantly showed an increased spread among the data sets. For the semi-annual variation, the amplitudes showed an increased spread, but for the phase the spread among the data sets was slightly reduced.

### 5.2.2 Different sampling in time and space

To derive the time series for the individual data sets the data were averaged in monthly and 10°latitude bins (see Sect. 3.1). The data coverage within these bins is not homogeneous in time and space for many data sets (Walker and Stiller, in preparation). This concerns in particular the occultation data sets. Other data sets provide global coverage on a daily basis but their measurement frequency is not daily, as for the MIPAS V5R MA (roughly every $10^{th}$ day), SCIAMACHY limb (globally every $8^{th}$ day, between 45°S and 45°N every $2^{nd}$ day) and SMR (varying throughout the mission, 10–15 days per month for the 544 GHz

data set and 4–8 days for the 489 GHz data set) data sets. All of these inhomogeneities can cause a sampling bias that influences the variability characteristics. Toohey et al. (2013) provided a thorough overview on this topic, focusing on sampling biases in monthly zonal mean climatologies (i.e. multi-year averages) of ozone and water vapour. They noted that none of the climatologies were completely free of a sampling bias, not even the MIPAS V5R NOM and MLS data sets which arguably had the densest sampling in time and space. They stressed the fact that the sampling bias is not only due to the actual sampling

pattern but additionally due the atmospheric variability within each time and latitude bin. In this regard, strong gradients across the polar vortex edge, hygropause or regions of dehydration as well as quick temporal changes, for example in the aftermath of SSWs, play an important role. One detail that we investigated was the influence of an incomplete coverage throughout a year on the variability characteristics. A typical example is the limited coverage of ACE-FTS in the tropics, providing only measurements in February, April, August and October. Using this example, we analysed the sensitivity of the annual variation

by subsampling the MIPAS V5R NOM and MLS data sets to these four months. The largest amplitude sensitivity was found in the Antarctic. In the lower stratosphere (related to key feature #3) and in the lower mesosphere (related to key feature #5) an increase of more than 0.25 ppmv was found. Contrary, in the middle stratosphere (related to key feature #4) a decrease of the amplitude (-0.2 ppmv) could be observed. In relative terms these changes amounted to a sensitivity between 50% and 100%. At other latitudes the changes were typically within ±0.05 ppmv, with some notable exceptions in the northern hemi-

sphere in the upper stratosphere and lower mesosphere. In this altitude range, between the equator and about 45°N, relative changes exceeded occasionally ±50%. The incomplete temporal coverage did also affect the phase. Differences of more than ±1 month could be predominantly observed in the upper stratosphere and lower mesosphere. Overall, differences in coverage can substantially contribute to differences in the variability characteristics derived from the data sets.

### 5.2.3 Temporal variations of systematic errors

All data sets have systematic errors. If these errors vary with time they can affect the variability characteristics derived from the data set. Systematic errors are commonly associated with uncertainties in the instrument characterisation as well as spectroscopic parameters. Some of the differences observed for the SMR data sets in this analysis can for example be attributed to systematic errors that vary in time. SMR is a heterodyne instrument in which the received signal $\nu_{RF}$, the radio frequency, is




converted down to a smaller intermediate frequency $\nu_{IF}$ before detection. For this conversion the received signal is combined with a constant frequency ($\nu_{LO}$) and the intermediate frequency is given by the following relationship: $\nu_{IF} = |\nu_{LO} - \nu_{RF}|$. Due the absolute condition the intermediate frequency holds the information of two frequency bands, i.e. the signal (primary) sideband and the image sideband. The image sideband is suppressed by interferometers but in particular for the 489 GHz data

sets the suppression is poor so that spectral information from the image sideband leaks into the detected signal. This sideband leakage appears to be influenced by the temperature inside SMR, which is not very well characterised. As the temperature has a significant seasonal cycle there is profound effect on the retrievals and thus on the derived result compared here. Another prominent example in this regard is the HIRDLS data. During the launch of the instrument a piece of plastic was dislodged blocking large parts of its aperture (Gille et al., 2008). Tremendous efforts have been undertaken to recover atmospheric signals

with the most complicated part being the characterisation of the blockage signal. Profile-to-profile comparisons within another WAVAS-II study have indicated a time variation systematic errors in particular above 30 hPa that affect the results of this study (Lossow et al., in preparation). Many more examples for the data sets involved in this comparison could be listed. Overall, temporal variation of systematic errors plays an important role for the differences among the data sets.

### 5.2.4 Differences in vertical resolution

The data sets considered in this work have different vertical resolutions. Full details are given in the WAVAS-II data set overview paper by Walker and Stiller (in preparation). The vertical structure of water vapour in the stratosphere and lower mesosphere is relatively smooth. Thus, generally only small effects due to differences in the vertical resolution are expected. Larger effects are only predicted in altitude ranges where the vertical water vapour distribution is more structured. This concerns the hygropause region in the lowermost stratosphere, the tape recorder region in the lower tropical stratosphere and, to some extent, the rela-

tively broad middle atmospheric maximum higher up. Hegglin et al. (2013) showed a comparison of the climatological seasonal cycle from various satellite data sets in the lowermost tropical stratosphere and found clear differences in amplitude and phase between the individual data sets in the vicinity of tropical hygropause. Subsequently, Weigel et al. (2016) discussed differences in the phase of the annual variation in the lower tropical stratosphere considering the SCIAMACHY limb and MLS v3.3/3.4 data sets. They found that these differences could be attributed to differences in the vertical resolution of the two data sets,

with the SCIAMACHY data set lower vertically resolved than the MLS v3.3/3.4 data set by a factor 1.5 to 4 (see also Walker and Stiller, in preparation). Another aspect are temporal variations of the vertical resolution in the vicinity of the hygropause which can affect the variability characteristics. In this regard, Schieferdecker et al. (2015) discussed the smaller annual cycle of MIPAS-IMKIAA data sets compared to the HALOE v19 data set. The corresponding MIPAS-IMKIAA retrievals did not consider volume mixing ratio directly as the unknown variable but its logarithm. This leads to stronger retrieval constraints during

the dry season accompanied by a lower vertical resolution. In the wet season, the conditions are reversed, thus explaining the annual variation in the altitude resolution and all of its consequences. Besides the MIPAS-IMKIAA data sets also other data sets considered here use a log retrieval approach and are thus similarly affected (Walker and Stiller, in preparation). In addition, we performed simple tests with model simulations to study the influence of the vertical resolution on the characteristics of the seasonal cycle. The high vertical resolution ($\sim$1 km) simulations were degraded with Gaussian kernels using different widths





to emulate different vertical resolutions. Subsequently the amplitudes and phases of the annual variation were derived. We focused on the tape recorder region and the middle stratosphere in the Antarctic where the middle atmospheric water vapour maximum can be found in wintertime. The tests indicated a clear influence of the vertical resolution in these regions, causing both amplitude variations and phase changes. Depending on the vertical resolution and altitude, variations of more than 50%

in amplitude and phase changes beyond $\pm 1$ month could be observed. Hence, in some regions, the vertical resolution can be an important factor in explaining differences in the variability characteristics derived from different data sets.

### 5.2.5 Influences of clouds and aerosols

The presence of clouds can introduce errors into the water vapour satellite retrievals. For this study, PSCs in the Antarctic during winter and spring are the most significant clouds impacting variability amplitudes and phases. There are obvious dif-

ferences in the magnitude of the dehydration due to these clouds that clearly influence the annual (key feature #3) and the semi-annual variation (key feature #2). Upper tropospheric clouds also contaminate retrievals and impacts can be seen in the lower stratosphere. This is on one hand due to the finite altitude resolution of the satellite data sets and on the other hand due to the propagation of errors. The measurement technique itself plays an important role for the cloud influence. Observations in the microwave region (as by MLS, SMILES or SMR) are less affected by clouds than observations in the infrared (e.g.

ACE-FTS, HALOE, MIPAS). The latter set of observations cannot measure in the presence of clouds above a given optical depth, hence the measurements will be biased towards cloud-free situations (i.e. biased dry). Similarly the observation geometry plays a role for the cloud influence. Observations by solar occultation (e.g. ACE-FTS, HALOE, SAGE) will be less influenced than lunar (SCIAMACHY lunar) and stellar (GOMOS) occultation measurements due to the much stronger measurement signal. The same relation is valid if comparing solar occultation to emission (particularly MIPAS) and solar scattering

(SCIAMACHY limb) measurements. Beyond that there are even differences in the cloud screening among the MIPAS data sets from the different processors. Hence, given the variety of measurement techniques and observation geometries the data sets used here are based on, differences are unavoidable in the regions where clouds exist and assert an influence. Since cloudiness varies over the year and is also affected by the QBO it impacts all modes of variability in water vapour addressed in this study.

In a similar way to the cloud influence described above the observations are affected by aerosols. This concerns primarily

the troposphere and lower stratosphere and the impact varies again among the data sets. On one end there is a data set like the SCIAMACHY limb data set for which clear effects from aerosols have been found (Weigel et al., 2016). On the other end the microwave observations of MLS, SMILES and SMR are rather insensitive to aerosols. The influence is especially profound after volcanic eruptions. Since the millennium there have been a number of large (i.e. volcanic explosivity index = 4) volcanic eruptions that had a major influence on the stratospheric aerosol loading (e.g. Vernier et al., 2011).

### 30 5.2.6 NLTE effects

NLTE effects are important in the upper stratosphere and the mesosphere, especially during daytime. The MIPAS data sets are influenced by these effects. However, with the exception of the MIPAS-IMKIAA V5R MA data set, none of them considers this influence explicitly (except by selecting spectral information unaffected by NLTE for the retrieval). Stiller et al. (2012a)



performed a sensitivity study for MIPAS-IMKIAA daytime water vapour data with and without considering the NLTE effects (their Fig. 15). Differences between the two cases could be observed above 30 km. At 55 km, retrievals without NLTE consideration led to 5% more water vapour than the NLTE retrieval. Above, a steep change occurred and at 65 km, retrievals without NLTE consideration led to 20% less water vapour compared to the NLTE retrieval. Due to the annual variation in insolation,

these NLTE effects will influence the characteristics of the annual and semi-annual variation derived from the MIPAS data sets which include a combination of day and nighttime data. During nighttime, NLTE effects are considerably smaller weakening the overall influence. To provide an estimate of the influence we recalculated the variability characteristics based on MIPAS nighttime data only (except for the MIPAS-IMKIAA V5R MA data set). The diurnal variation of water vapour in the altitude range of interest is generally small (Haefele et al., 2008) and, thus, should not be a decisive factor. In the discussion we focus

only at the latitude range between 60°S and 60°N. At higher latitudes the results are also influenced by the missing coverage of the nighttime data during summer. The amplitudes were most affected in the lower mesosphere, in particular above 0.4 hPa. Here, the changes were largest for the annual (ranging from -0.4 ppmv to 1.2 ppmv) and semi-annual variation (ranging from -0.2 ppmv to 0.5 ppmv). The large positive changes occurred between 60°S and 30°S as well 15°N and 45°N. For the QBO variation the changes were typically within ±0.1 ppmv. The agreement of the MIPAS data sets improved polewards 45°for the

annual variation while for the semi-annual variation a clear deterioration was found polewards of 30°. For the QBO variation varying results were found in this regard. Also in the upper stratosphere prominent positive changes could be observed for the annual variation, accompanied with a substantial decrease of the agreement among the MIPAS data sets. These changes occurred around 3 hPa to 2 hPa in the latitude ranges 60°S – 30°S and 25°N – 40°N. In terms of the phase the largest changes for all variability patterns also occurred in the lower mesosphere. In addition, larger changes were found in the upper strato-

sphere at mid-latitudes for the annual variation. As for the amplitudes, both improvements and degradations of the agreement among the MIPAS data sets have been found. The semi-annual and QBO variation showed prominent improvements in the lower mesosphere between 15°N and 45°N. For the annual variation such improvement was more visible at low latitudes, i.e. from 15°S to 15°N. On the other hand the differences among the MIPAS data sets significantly increased polewards of 35°N in the upper stratosphere for the annual variation. Combining the MIPAS nighttime results with the other data sets yielded

similar changes in the amplitudes, phases and the agreement of the data sets. Overall, it can be concluded that NLTE effects clearly affect the results. Minimising these effects by using only nighttime data, however does not always reduce the differences between the data sets.

## 5.3 Sensitivity of results based on approach

There are many ways on which the characteristics of the different variability patterns can be derived. This implies a sensitivity

of the results based on the chosen approach, as pointed out already in the approach section (Sect. 3.2). This may be an important aspect to be considered when comparing these results to other studies. Thus this section aims to quantify this sensitivity by considering a small set of other possible approaches. As in Sect. 5.2 changes are given as sensitivity test results minus the results presented in Sect. 4.





The regression models for the semi-annual (Eq. 4) and QBO (Eq. 5) variation provided also estimates of the annual variation. Those results were used to estimate the sensitivity of the annual variation depending on the derivation approach. Often the regression model for the semi-annual variation yielded similar results for the annual variation as the regression model (Eq. 1) specifically used for the analysis of the annual variation. The amplitudes were generally within ±0.05 ppmv or about ±10% in relative terms. The high latitudes were the only exception. In the Arctic in the middle and upper stratosphere, changes of up to 0.1 ppmv were visible. In the Antarctic in the lower stratosphere (key feature #3) the amplitude differences exceeded 0.3 ppmv. Here also the phase differences are larger than a month, i.e. the annual maximum is observed later in the regression model for the semi-annual variation. In the middle stratosphere (key feature #4) changes of -0.1 ppmv were found while in the upper stratosphere the changes were of the same size but positive again. The results from the regression model for the QBO variation closely resembled those from the regression model for the semi-annual variation. Hence the inclusion of the QBO terms into the regression model did not impose any additional sensitivity for the variability characteristics of the annual variation.

Another common approach to derive the characteristics of the annual variation is based on multi-year average (climatological) distributions, typically considering monthly zonal means. From this set of data, the amplitudes and phases can be determined via the size and the time of the annual extrema. To be consistent with the results from the regression approach the amplitude of the annual variation is given by:

$$\mathbf{A}_{AO,climatology} = 0.5 \cdot (\mathbf{VMR}_{max} - \mathbf{VMR}_{min}), \tag{8}$$

where $\mathbf{VMR}_{max}$ and $\mathbf{VMR}_{min}$ denote the maximum and minimum water vapour volume mixing ratios during the year. The phase is again defined by the month in which the annual maximum occurs. Figure 15 shows an example comparison between the two approaches using the MIPAS-ESA V5R NOM data set. The results from the climatology approach are shown in the top row. The data handling for the climatology data set followed the same path as used for the time series data, except for the temporal binning (see Sect. 3.1). The middle row shows the results for the original time series approach and the lower row shows the difference between both approaches. The latitude-altitude distribution for the amplitude is the same, but obvious quantitative differences are visible. In most areas, the climatology approach exhibits larger amplitudes, consistent with the smoothing character of the sinusoids used in the regression. The largest differences in that direction can be observed at the high latitudes, the tropical upper stratosphere and the southern hemispheric lower mesosphere. Here the differences frequently exceed 0.25 ppmv and reach even 1 ppmv. In relative terms, this amounts to a wide range, up to several hundred percent. Compared to the time series approach significantly smaller amplitudes in the climatology approach are only found in the Antarctic at 10 hPa, around 20°S and 2.5 hPa and polewards of 45°N around 0.15 hPa. Also in the phase considerable changes between the two approaches are evident. Given that the regression model does not use any overtones of the annual cycle some uncertainty in the exact position of annual extrema is reasonable. The bulk of the differences are negative indicating that the annual maximum is found earlier in the climatology approach. Generally the differences are within ±2–3 months. The largest absolute changes occur in the lower stratosphere at high latitudes and in the tropical and subtropical upper stratosphere, which shows both negative and positive changes. The example differences between the climatology and regression approach given



in Fig. 15 are representative of the situation when all data sets are considered. Using the climatology approach instead of the regression approach considerably increases the spread among the data sets and thus enhances the uncertainties in the variability characteristics.

To test the sensitivity of the approach for the semi-annual variation, the results obtained with the regression model for the

QBO variation (Eq. 5) were employed. In terms of the amplitude the two sets of results exhibited only small differences. In relative terms somewhat more pronounced changes could be observed in the lower stratosphere, ranging from -20% to 40%. For the phase, the changes were typically within $\pm0.2$ months.

For the sensitivity of the QBO variation, a test was performed where the regression model used the normalised Singapore winds at 30 hPa and 10 hPa as QBO proxies (instead of 50 hPa and 30 hPa as in Eq. 5). The winds at 30 hPa and 10 hPa are

also approximately orthogonal and have been used in other studies (e.g. Kyrölä et al., 2013). For the amplitude, the majority of changes were small. The largest changes (>0.05 ppmv) occurred in the areas that were marked as key features in Fig. 11. In relative terms, the largest changes were found in the tropical upper stratosphere (key feature #2) and in the Arctic in the middle and upper stratosphere (key feature #3). Those changes were positive and as large as 50%. The phase changes were typically within $\pm4$ months, maximising in the upper tropical stratosphere (key feature #2).

In summary, the sensitivity studies showed that the amplitudes of the different variability patterns were affected by the chosen approach, but the typical latitude-altitude distributions remained rather consistent. The phases were more sensitive to the approach, both the overall distribution and the actual values were affected. In general, the most sensitive region was the polar region. This result needs to be kept in mind when comparing the variability characteristics based on different approaches.

## 6 Conclusions

Satellite data sets of water vapour in the stratosphere and lower mesosphere were compared with respect to the amplitude and phase of the annual, semi-annual and QBO variation. The comparisons indicated a rather consistent picture of the latitude-altitude distribution of the amplitudes of the three variability patterns. Quantitatively, there were however obvious differences in the amplitudes derived from the individual data sets. Larger differences were typically observed at high latitudes, close to the tropopause and in the lower mesosphere. At low latitudes larger differences were found for the QBO variation. Depending

on variability pattern, latitude and altitude the spread among the data sets amounted to several tenths of 1 ppmv, in extreme cases they even exceeded 1 ppmv.

In the problematic regions, the standard deviation over all data sets exceeded 0.2 ppmv for the annual variation. For the two other variability patterns, standard deviations exceeding 0.1 ppmv were found to be characteristic for these regions. The relative standard deviation (using the average amplitude over all data sets as reference) revealed a different picture. Many

regions with large spreads in the amplitude showed relatively small relative standard deviations (<50%) as the variability in those is typically large. Regions with small spreads in amplitude often coincided with low variability resulting in larger relative standard deviations (>50%). For the phase, the lowest uncertainties were typically found in regions where the variability is large. In the lower mesosphere larger differences between the data sets could be observed. In general the phase differences



were within $\pm 1$ month for all three variability patterns. The standard deviations over all data sets were found to be typically smaller than a month for the annual and semi-annual and smaller than 5 months for the QBO variation.

There are multiple reasons that give rise to the observed differences between the individual data sets. Important contributions arise from temporal variations of systematic errors and differences in the temporal and spatial sampling. Besides that,

different time periods (especially important for the QBO variation), the vertical resolution of the data or the influence of clouds, aerosols and NLTE effects also contribute to the differences. There are various ways to derive the characteristics of the different variability patterns. Different approaches lead to quantitative differences in the amplitude and phase estimates, which need to be considered in comparisons with other results. The latitude-altitude distribution of the amplitude is quite insensitive to the derivation approach while the phase is more sensitive.

Overall the results provide valuable constraints on the characteristics of shorter term variations in stratospheric and lower mesospheric water vapour for subsequent modelling or observational studies. A prominent example is the simulation of the atmospheric tape recorder that still exhibit fundamental differences to the observations, both with respect to the amplitude but especially the phase (e.g. Eyring et al., 2006).

*Acknowledgements.* The Atmospheric Chemistry Experiment (ACE), also known as SCISAT, is a Canadian-led mission mainly supported

by the Canadian Space Agency and the Natural Sciences and Engineering Research Council of Canada. We would like to thank the European Space Agency (ESA) for making the MIPAS level-1b data set available. SCIAMACHY spectral data have been provided by ESA. The work on the SCIAMACHY water vapour data products has been funded by DLR (German Aerospace Center) and the University of Bremen. The SCIAMACHY limb water vapour data set v3.01 is a result of the DFG (German Research Council) Research Unit "Stratospheric Change and its Role for Climate Prediction" (SHARP) and the ESA SPIN (ESA SPARC Initiative) project and were partly calculated using

resources of the German HLRN (High-Performance Computer Center North). S. Lossow was also funded by the SHARP project under contract STI 210/9-2. We thank SPARC and WCRP (World Climate Research Programme) for their guidance, sponsorship and support of the WAVAS-II programme.





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



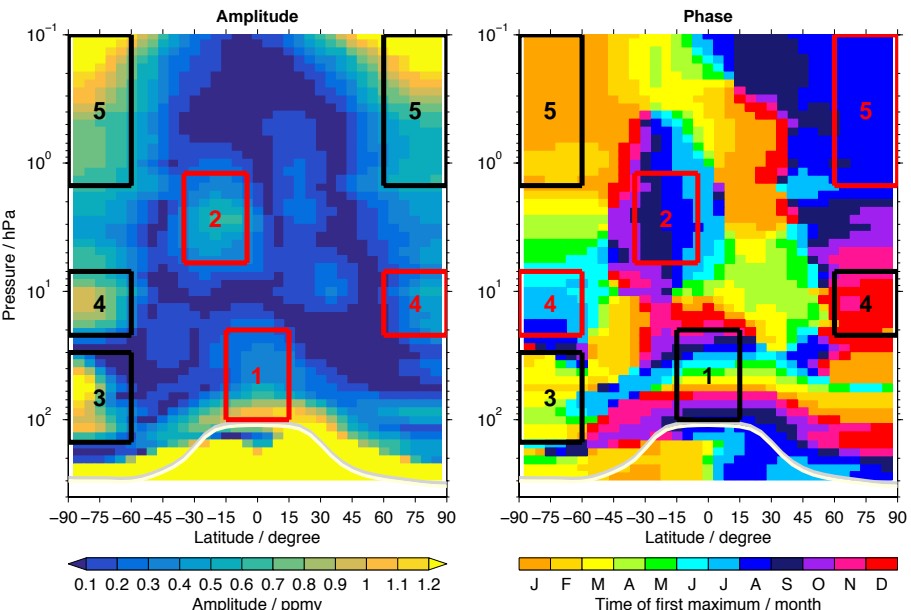

**Figure 1.** Example of the annual variation characteristics as function of latitude and altitude based on the MLS v3.3/3.4 data set. The left panel shows the amplitude and in the right panel the phase is shown. The phase is represented by the month in which water vapour exhibits its annual maximum in the regression fit. The light grey and white line indicates the mean tropopause (2000 – 2014) as derived from the MERRA reanalysis data. The red and black boxes mark characteristic features of the annual variation in water vapour that are described in more detail in the text. The colour variation of the boxes is for better contrast. White areas indicate that there is no data.





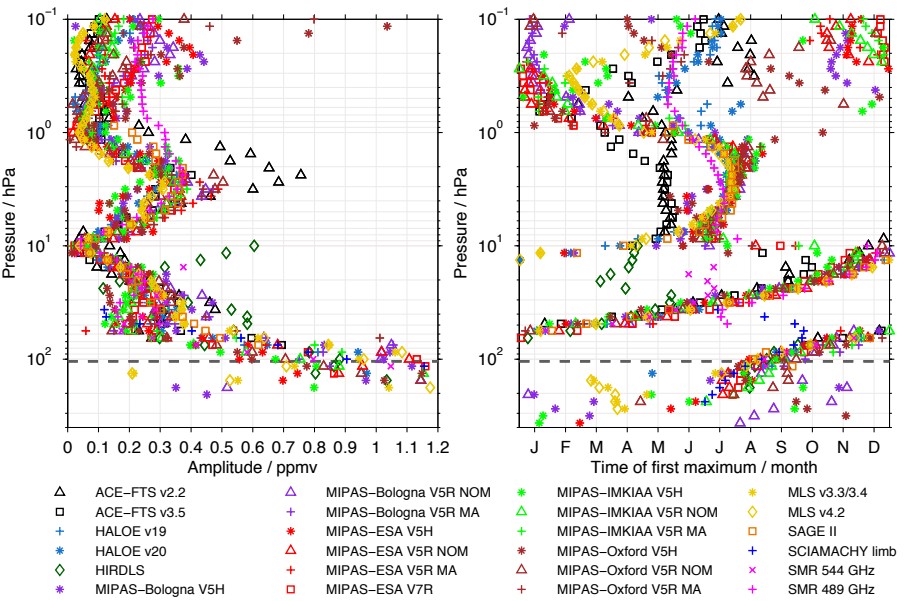

**Figure 2.** The annual variation in the inner tropics ($5°S - 5°N$) as function of altitude as seen by the different data sets. Like in Fig. 1 the left panel shows the amplitude of the variation while the right panel shows the phase. The dark grey dashed line indicates the mean tropopause for the latitude band, again using MERRA data for the time period $2000 - 2014$. The month ticks in the right panel refer to the centres of the individual months.

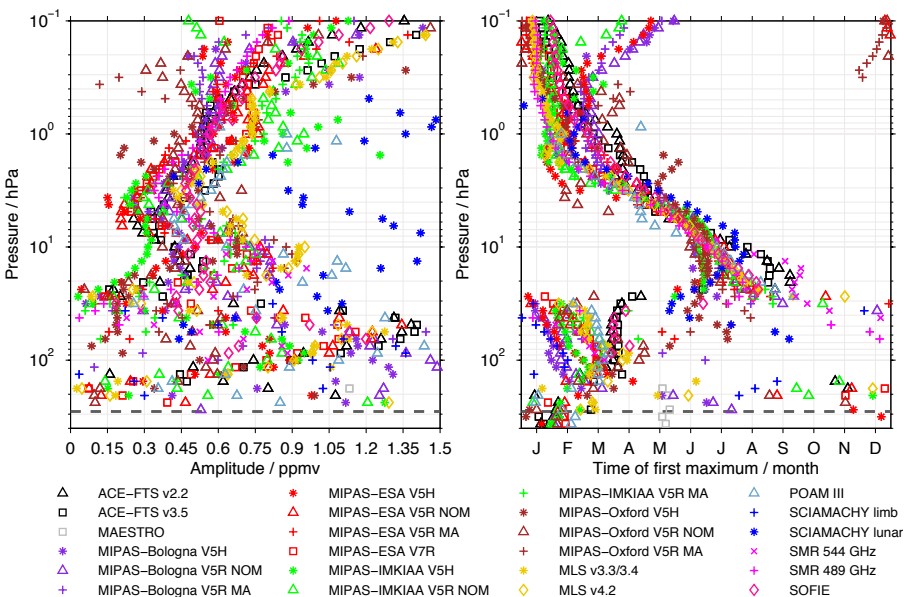

**Figure 3.** As Fig. 2 but here for the latitude band from $85°S$ to $75°S$. Note that the scale of the amplitude axis is extended compared to the previous figure to accommodate the larger spread among different data sets.





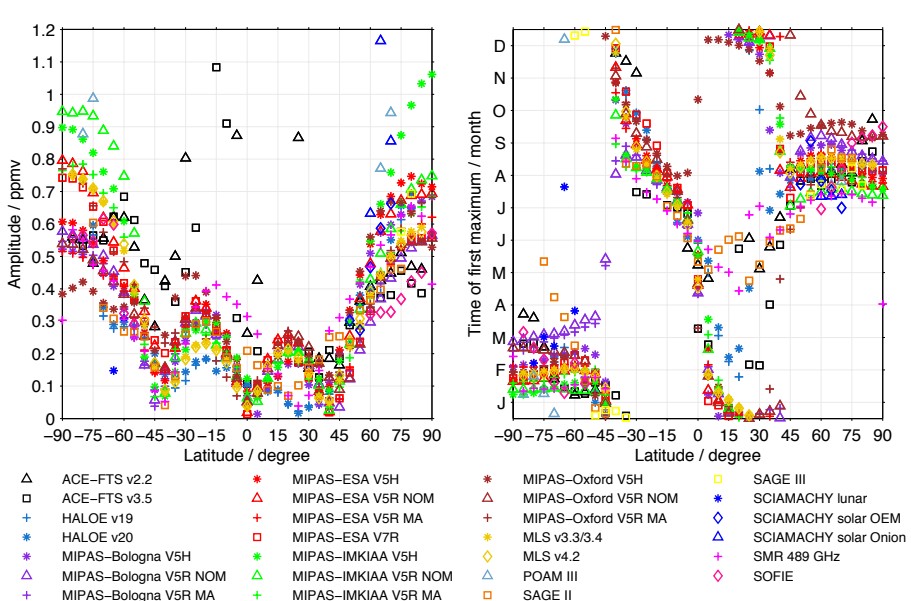

**Figure 4.** The annual variation as observed by the individual data sets at the 1 hPa pressure level over the entire latitude range.







**Figure 5.** The typical uncertainties among the different data sets for the amplitude and phase of the annual variation. The upper panel shows the standard deviation for the amplitude in absolute terms $\sigma(\mathbf{A})$ while the middle panel considers the deviation in relative terms $\sigma_{rel}(\mathbf{A})$ (with respect to the average over all data sets, see Eq. 6). The lower panel shows the standard deviation of the phase difference $\sigma(\Delta\mathbf{P})$ with respect of the MLS v4.2 data set. Some data screening has been applied to achieve meaningful results, see text for details. As in Fig. 1, the light grey and white line indicates the mean MERRA tropopause for the time period 2000 – 2014.




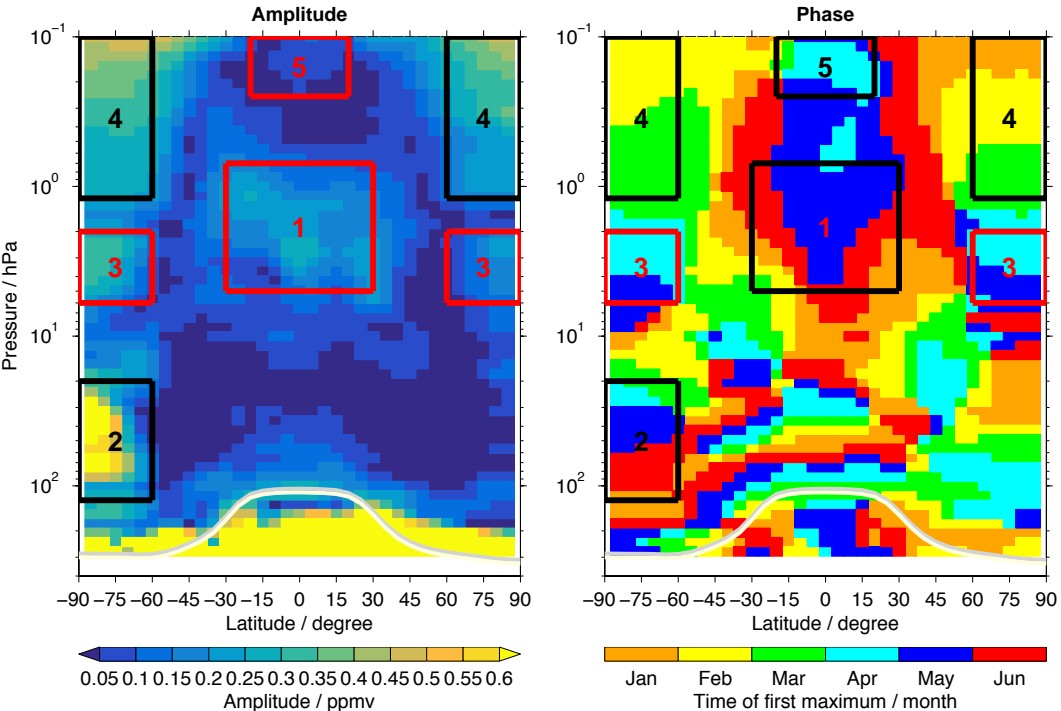

**Figure 6.** As Fig. 1 but here for the semi-annual variation. Note that the range of amplitudes is smaller here. The phase denotes the month of a calendar year were the first water vapour maximum is found. The example is again based on the MLS v3.3/3.4 data set.




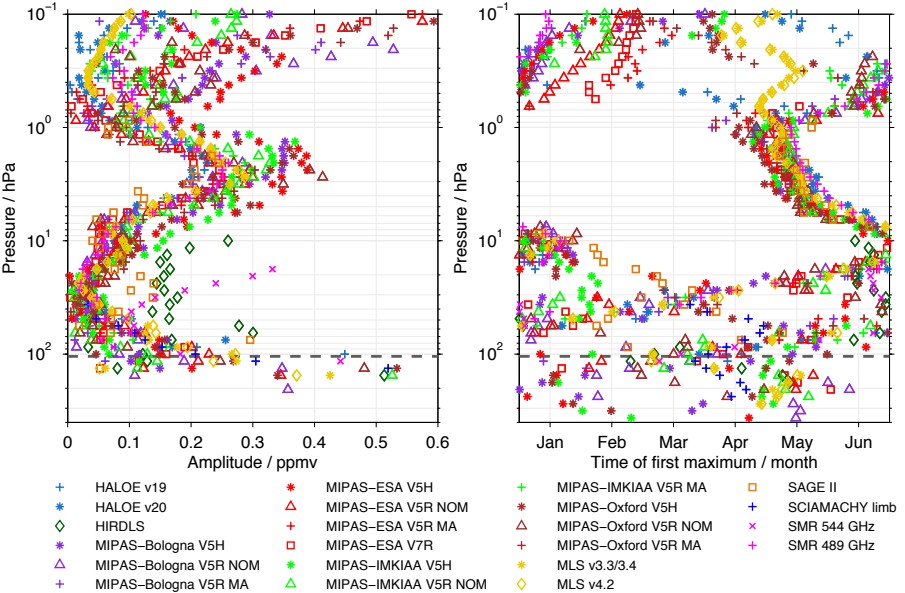

**Figure 7.** The characteristics of the semi-annual variation in the inner tropics (5°S – 5°N) as derived from the individual data sets. The left panel shows the amplitude of the semi-annual variation, the right panel shows the phase in terms of the month in which the first water vapour maximum occurs during a calendar year. The average tropopause is marked by the dark grey dashed line.

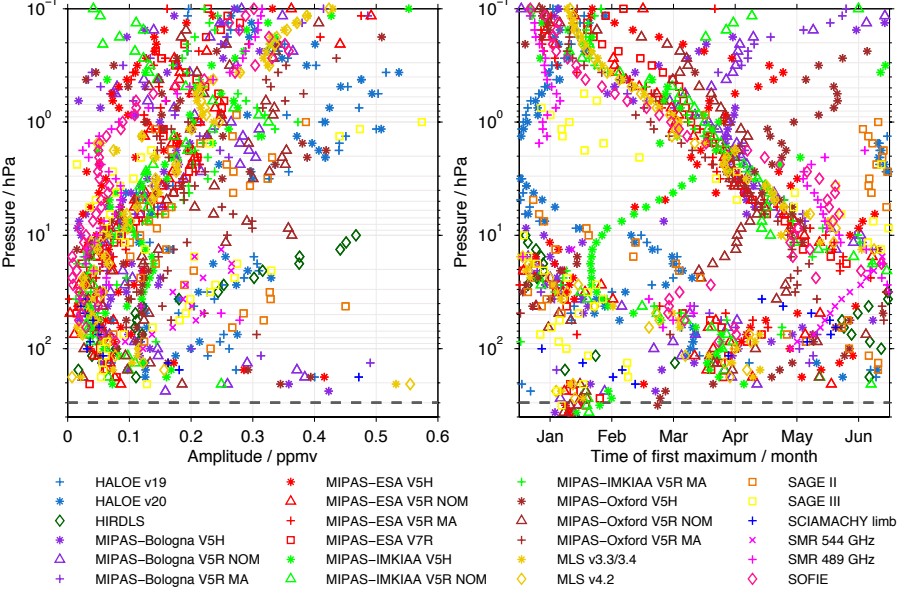

**Figure 8.** As Fig. 7 but here the results for the latitude range between 70°N and 80°N are shown.





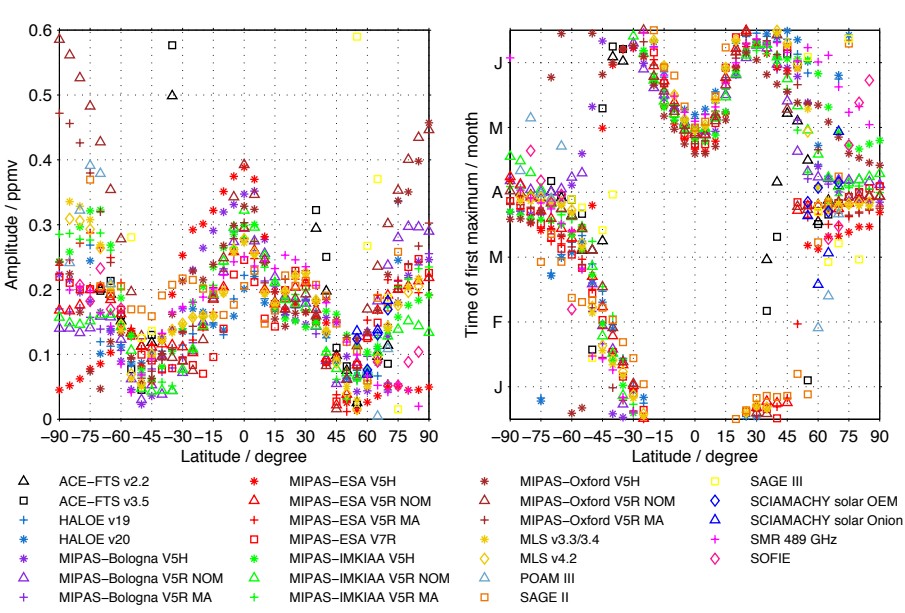

**Figure 9.** The latitude distribution of the semi-annual variation at 2.4 hPa, close to the pressure level where the tropical semi-annual variation peaks as shown in Fig. 6 and 7.





**Figure 10.** As Fig. 5 but here the semi-annual variation is considered. Note the different scale ranges for the absolute standard deviation in amplitude and phase compared to Fig. 5.





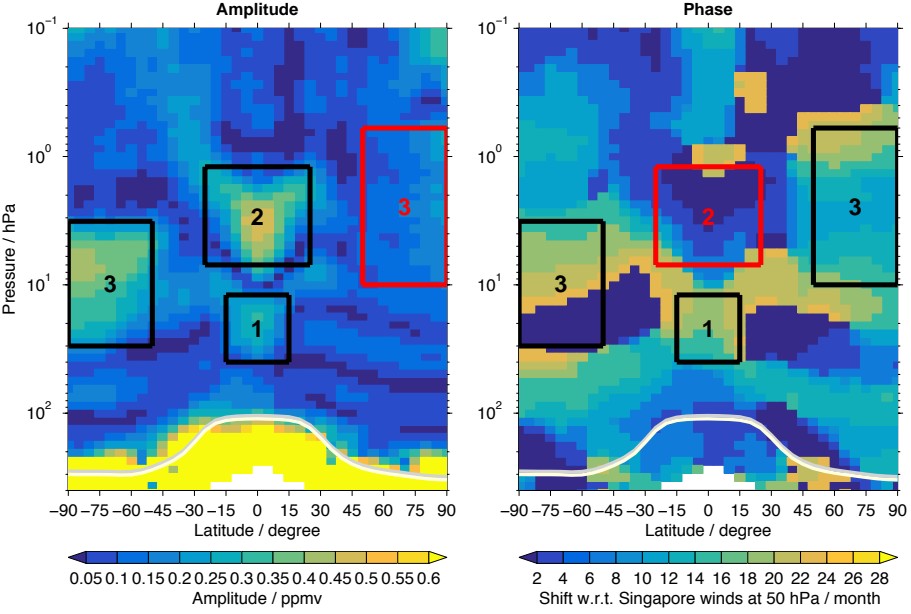

**Figure 11.** Example of the variability characteristics for the QBO variation based on the MIPAS-IMKIAA V5R NOM data set. The phase is derived as the shift of the QBO regression fit for which the correlation with the Singapore (1°N, 104°E) winds at 50 hPa maximises.

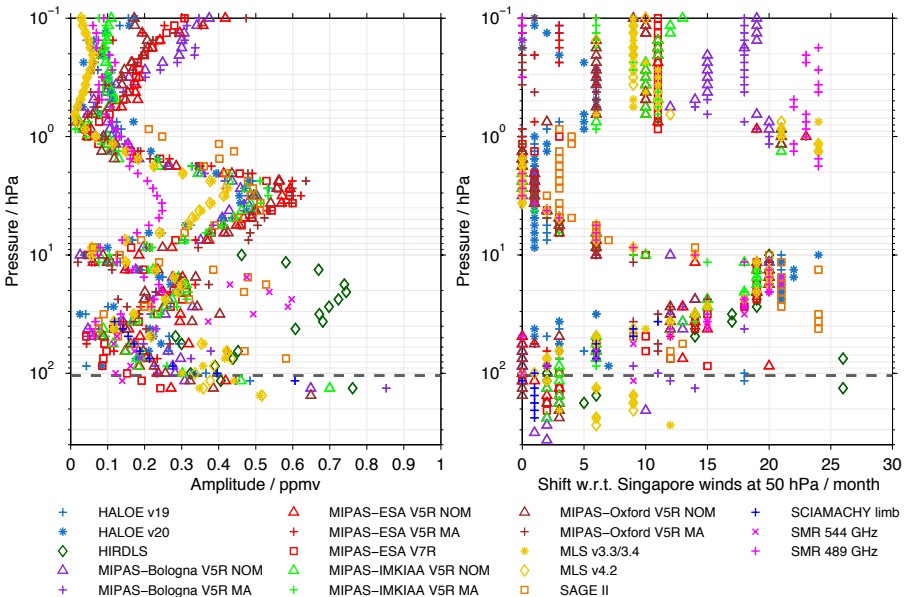

**Figure 12.** The QBO variation in the tropics (5°S − 5°N) as derived from the different data sets.





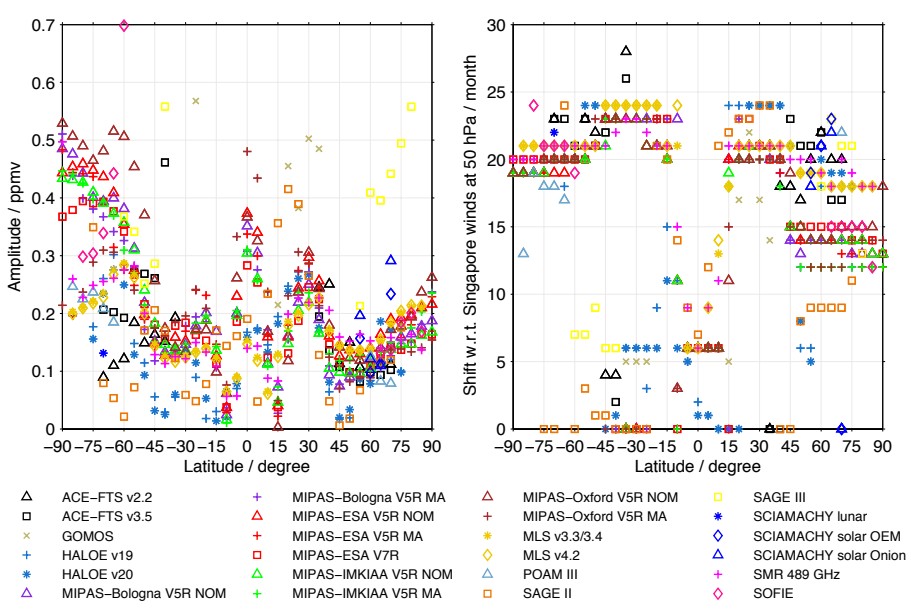

**Figure 13.** The latitude distribution of the QBO variation at a pressure level of 7.5 hPa.




**Figure 14.** As Fig. 5 and 10 but here focusing on the QBO variation.





**Figure 15.** Comparison of amplitude and phase for the annual variation as derived with the climatology approach (top row, see Sect. 5.3) and the time series regression approach (middle row, see Sect. 3.2). The bottom row shows the differences between the two approaches. This example is based on results of the MIPAS-ESA V5R NOM data set.





**Table 1.** Overview over the water vapour data sets from satellites used in this study.

| Instrument | Data set | Labeled as | Time period |
|---|---|---|---|
| ACE-FTS | v2.2 | ACE-FTS v2.2 | 02/2004 – 09/2010 |
| | v3.5 | ACE-FTS v3.5 | 02/2004 – 03/2013 |
| GOMOS | LATMOS v6 | GOMOS | 09/2002 – 09/2011 |
| HALOE | v19 | HALOE v19 | 10/1991 – 11/2005 |
| | v20 | HALOE v20 | 10/1991 – 11/2005 |
| HIRDLS | v7 | HIRDLS | 01/2005 – 03/2008 |
| MAESTRO | Research version | MAESTRO | 02/2004 – 03/2013 |
| MIPAS | Bologna V5H v2.3 NOM | MIPAS-Bologna V5H | 07/2002 – 03/2004 |
| | Bologna V5R v2.3 NOM | MIPAS-Bologna V5R NOM | 01/2005 – 04/2012 |
| | Bologna V5R v2.3 MA | MIPAS-Bologna V5R MA | 01/2005 – 04/2012 |
| | ESA V5H v6 NOM | MIPAS-ESA V5H | 07/2002 – 03/2004 |
| | ESA V5R v6 NOM | MIPAS-ESA V5R NOM | 01/2005 – 04/2012 |
| | ESA V5R v6 MA | MIPAS-ESA V5R MA | 01/2005 – 04/2012 |
| | ESA V7R v7 NOM | MIPAS-ESA V7R | 02/2005 – 03/2012 |
| | IMKIAA V5H v20 NOM | MIPAS-IMKIAA V5H | 07/2002 – 03/2004 |
| | IMKIAA V5R v220/221 NOM | MIPAS-IMKIAA V5R NOM | 01/2005 – 04/2012 |
| | IMKIAA V5R v522 MA | MIPAS-IMKIAA V5R MA | 01/2005 – 04/2012 |
| | Oxford V5H v1.30 NOM | MIPAS-Oxford V5H | 07/2002 – 03/2004 |
| | Oxford V5R v1.30 NOM | MIPAS-Oxford V5R NOM | 01/2005 – 04/2012 |
| | Oxford V5R v1.30 MA | MIPAS-Oxford V5R MA | 01/2005 – 04/2012 |
| MLS | v3.3/3.4 | MLS v3.3/3.4 | 08/2004 – 12/2014 |
| | v4.2 | MLS v4.2 | 08/2004 – 12/2014 |
| POAM III | v4 | POAM III | 04/1998 – 12/2005 |
| SAGE II | v7.00 | SAGE II | 10/1984 – 08/2005 |
| SAGE III | Solar occultation v4 | SAGE III | 03/2002 – 11/2005 |
| SCIAMACHY | Limb v3.01 | SCIAMACHY limb | 08/2002 – 04/2012 |
| | Lunar occultation v1.0 | SCIAMACHY lunar | 04/2003 – 04/2012 |
| | Solar occultation - OEM v1.0 | SCIAMACHY solar OEM | 08/2002 – 04/2012 |
| | Solar occultation - Onion peeling v4.2.1 | SCIAMACHY solar Onion | 08/2002 – 04/2012 |
| SMR | v2.0 544 GHz | SMR 544 GHz | 11/2001 – 12/2014 |
| | v2.1 489 GHz | SMR 489 GHz | 11/2001 – 08/2014 |
| SOFIE | v1.3 | SOFIE | 05/2007 – 12/2014 |