# Peer review of "The SPARC water vapour assessment II"

_Atmospheric Measurement Techniques, 2016_

## Referee Comment (RC1) · H. C. Pumphrey (Referee) · 9 Dec 2016

**General remarks**

The paper presents a very careful and detailed description of the annual, semiannual and quasi-biennial variations in middle-atmosphere water vapour, as observed by a large suite of satellite instruments. It is a very useful summary and should be published subject to some minor and technical corrections.

The written English is mostly very good. It has signs of being written by a non-native

speaker (which is fine) but also has a few sentences which make no sense; these should be fixed. The writer has a tendency towards over-long sentences with inadequate punctuation; it would benefit from an attempt to improve the wording in these cases.

The figures are produced to a good standard and are very much improved from the pre-discussion technical review. I note that the zonal mean plots (e.g. Figure 1) appear in a wide variety of aspect ratios. In Figure 1 the plots are taller than they are wide, in Figure 5 (and much of the appendix) they are wider than they are tall, and in Figure 15 they are almost square. This makes it harder than it needs to be for the reader to compare one figure with another.

The typesetting of equations is not very good, especially considering that it was done with LaTeX. In particular:

- You do not need a dot for "multiply" and especially not between 2, $\pi$ and $t$.

- Function names sin, cos, tan, atan, etc. should be set in upright font. Use `\sin`, `\cos` etc. in LaTeX and `\mathrm{fnord}` for a function fnord which is too obscure for LaTeX to know about.

- It is not made clear why a number of quantities are typeset in bold.

- It is not good style to use long words like "offset" and "linear" as subscripts. If you insist on doing it, they too should be in upright font: `C_\mathrm{offset}` to give $C_{\mathrm{offset}}$.

**Specific corrections**

- Page 2, Lines 3–4: This sentence either makes no sense or the word order is very strange.

- Page 2 Line 12: The AMT house style is to write $0.3\,\mathrm{W\,m^{-2}}$, not $0.3\,\mathrm{W/m^2}$.

- Page 3, Line 2: Between "dehydration" and "the" strikes me as a good place to use a colon (:).

- Page 3, Line 21: Space should be after comma, not before it.

- Page 3, Lines 23–24: This sentence is particularly oddly worded and should be re-written.

- Page 5, Line 18. The use of "Be reminded" seems odd. Maybe replace with "Note"?

- Page 8, Line 4 and Page 15, Line 21: In both cases, "likely" should be replaced with "probably". Despite ending in "ly", the word "likely" is an adjective (synonymous with "probable") and should not be used as an adverb.

- Page 9 Lines 1–2: Box 2 in Figure 1 is in the middle and *upper* stratosphere, not the middle and *lower* stratosphere.

- Page 11, Line 21: "reveals" should read "is revealed".

---

## Short Comment (SC1) · 14 Dec 2016

WAVAS II is an updated evaluation of the quality of available water vapor data sets for the middle atmosphere. I contributed the historic Nimbus 7 LIMS Version 5 data set to the first SPARC WAVAS report of 2000. After that time, a LIMS Version 6 data set was created and evaluated and made available for use by the research community, and this newer version has some subtle, but important improvements for the overall distributions of its stratospheric water vapor. Those improvements are reported in Remsberg et al. (2009), in addition to the initial findings from LIMS version 5 that you cite (Remsberg

et al., 1984). Although you do not consider the LIMS data set for your current WAVAS II assessment, I respectfully request that you include the citation to Remsberg et al. (2009) on page 3, line 26 of your WAVAS II manuscript in place of Remsberg et al. (1984).

Remsberg, E. E., M. Natarajan, G. S. Lingenfelser, R. E. Thompson, B. T. Marshall, and L. L. Gordley, On the quality of the Nimbus 7 LIMS version 6 water vapor profiles and distributions, Atmos. Chem. Phys., 9, 9155-9167, doi:10.5194/acp-9-9155-2009, 2009.

---

## Referee Comment (RC2) · Anonymous Referee #2 · 17 Dec 2016

**General Comments**

This work describes the annual, semi-annual, and quasi-biennial (QBO) variability of water vapour, as calculated from a large number of satellite-retrieved data sets from 1984 to 2014, with some differences arising naturally from the different time periods considered. Other differences in the behaviours are probably the result of a combination of effects, including sampling issues, systematic but non-constant error sources, vertical resolution differences, clouds, aerosls, and non-local thermodynamic effects on some retrievals, among other possible issues. Overall, the analyses are performed in a

way that is sound and the paper is well written overall, but a few language adjustments and corrections are needed.

Some of my concerns and criticism relate to the legibility of some of the Figures, or some of the points shown in these Figures, as a result of the large number of data sets that were used in this comprehensive analysis and summary. Also, having the representation of so many MIPAS retrievals is somewhat distracting in terms of the end results which should state whether some instruments are clearly showing differences versus others, rather than too much of retrieval A versus retrievals B, C, D,... for the same instrument. An alternative approach would be to try to clarify the MIPAS results and summarise them by using some average result, although there may be too many ways to decide how to do this, which is why showing everything in a sort of politically correct way may still be the "best way". One could envision simply throwing out some of the outliers among the MIPAS results to show maybe only one or two (for the two main MIPAS periods) "cleaner" (or average/median) MIPAS results, and then discussing or showing more clearly how this compares to other satellite instruments/results. It is somewhat disconcerting to see that sometimes the range of results is quite large, but as this is the reality, we also need to better understand the limitations/errors involved, which also includes errors from different retrievals. Such retrieval errors would also exist if one had several groups working on other instrument data, although the actual spread of results is unknown. In the end, it is clear that the same atmosphere is being observed by all these instruments, but this spread in results does make it somewhat challenging for a potential comparison to model results (not for this paper itself); how would a modeler go about chosing what to compare to (brief suggestions are certainly welcome)? In certain sensitivity studies, it would also help to try to clarify what may be able to specifically explain some of the differences between the results (in particular, regarding the sampling). Also, certain differences are probably not significant; by that I mean that since the results are obtained through regression fits, there should also be a way to obtain uncertainties in these results. For example, in places where the amplitudes (or phases) are small, the results may well not be significant, and/or

the differences between the various results are not significant. In theory at least, this could be quantified better, so that not so much time is spent worrying about certain differences in the end. As this could represent a fairly large amount of work, this is just a suggestion to look into. Also, it would be better if this paper could refer back to some of the references listed (for example on page 3, lines 26/27) in terms of how the current results might differ, or agree, with previous findings. Finally, using the present tense more abundantly would be useful in my opinion, and trying to make some data sets more visible could also help, although this is clearly a challenge, if the number of points is not reduced (I also imagine that a large number of colored lines is not really a viable solution).

After the above items are addressed as well as possible (without needing to embark on a large amount of extra work), this would definitely be a useful paper to see published. The paper's usefulness does depend on its clarity regarding the final results and potential explanations for some of the larger differences.

Specifics

- I recommend that you use the present tense in most of the paper (just a suggestion); I find that there mixture of past and present could be improved upon.

- The Abstract values for amplitudes would be useful as percentages, also probably in the Conclusion section (not just 0.2 ppmv, 0.1 ppmv).

- P1, L3, probably worth adding "vapour" after "water", and also on L5. - L9, "In these regions, the standard deviation over all data sets..." - L10, "For the phase, the larger differences between data sets are [or were] found in the lower mesosphere."

- P2, L1, "The standard deviations of the phases for all data sets are [were] typically ..." - L2, The amplitude and phase differences among the data sets are probably caused by a combination of factors, including differences in temporal and spatial sampling, and temporal variations in systematic [retrieval?] errors. I would note that it would be

helpful if you were able to point to the largest likely sources of such differences, given the work that has been done already, although it seems that this may be challenging to converge upon. - L10, add a comma after "greenhouse gas". - L20, change "effect" to "affect".

- P3, L23-24, Reword this, e.g. "A complete understanding of water vapour changes requires a good description of annual, semi-annual, and quasi-biennial oscillation variations (denoted here as QBO variation)." - L25, "shorter-term". - L33, "QBO variations which are subsequently summarised".

- P4, L16, "referred to the WAVAS-II...".

- P5, L3, "which we collectively..." - L20, maybe putting the "1 + " at the front of this equation would make it clearer.

- P6, L4, "with the phase denoting the time of maximum in the semi-annual fit between...". - L23, add a comma after "expected".

- P7, L5, change "Additionally" to "In addition". - L22, "screened amplitudes"

- P8, L1, The largest uncertainties were found in the lower mesosphere. For the reference data sets, we considered those that have a more or less... - L10, such plots are provided for all data sets considered in this work. - L11, allowing for a direct comparison. - L12, Finally, a summary is provided in the form of...in Sect. 3.3. - L20, vapour as a function of...

- P9, L2, I think you mean upper stratosphere (not sure why middle really). Also, late summer and early autumn would apply for Aug-Sep, which is really what you should state, since you are referring to the SH, where summer and autumn do not occur in Aug-Sep, I would say. - L5, allows for more effective downwelling of moister air from above, these mositer values arising from methane photochemistry. - L12/13, occurs at a rather constant level... but more of a seasonal characteristic. - L20, except that feature #5 occurs at altitudes above the water vapour maximum. During winter,... -

L26, add a comma after "coverage", and also after "data sets)", and say "or sometimes only a subset of those." - L31, as a function of altitude.

- P10, L16, "on the order of" [please change this everywhere, e.g. on L26, L35 too] - L19, data sets did not have sufficient temporal coverage... - L25, 20 hPa, good agreement ... - L29, deviate in obvious ways from...

- P11, L9, while the majority of data sets - L14, in the form of - L20, is revealed [not reveals] - L20, The former lie typically in regions where

- P12, L4, Note that the amplitude - L32, a breakup of the vortex breakup sounds a bit strange. Do you mean an interruption of the vortex breakup?

- P13, L9, SSAO allows waves to propagate further up only if they have horizontal propagation directions opposite to the zonal wind...

- P14 - Fig. 9 is an example where SAGE III data, in yellow, are very hard to read/find even on a screen view of the plot(s), and on a printed version too. - L30, do you mean "despite the small absolute standard deviations" or "as a result of the small absolute standard deviations"?

- P15, L2, delete "above". - L16, adding a bit more clarification regarding the "profound effect" [maybe from the Jackson ref.] would be useful. - L22, add a comma after "feature #1". - L31, This is also a characteristic of key feature #2...

- P16, L26, there are a few data sets that...

- P17, L2, add a comma after "patterns". - L14, change "lowest" to "poorest". - Summary, here (again), it would sound nicer if the present tense could be used. - L26, add a comma after "patterns" - L29, add a comma after "the key features" - L30, add "and at" before "high latitudes"

- P18, L2, 86%, and considering a standard deviation... - L4, change "could" to "can" or just "are often observed".

- P19, L5, add a comma after data sets. - L14, sampling bias is due not only to the actual... - L15, bu talso to the atmospheric variability... - L17, We investigated the influence of incomplete coverage throughout the year. - L22, Contrary to this, in the middle stratosphere... and change "could be" to "was" or "is". - L26, changes exceed[ed] 50% on occasion. - Also, did the sub-sampled data sets help to \*improve\* the comparisons in these sensitivity test, it is not clear what the result really is [it would make sense if it did, otherwise, there are other factors that cause the differences]?

- P20, L3, The intermediate frequency holds the information... - L11, a time variation of some systematic errors (in partcular above 30 hPa) that affects the results - L22, in the vicinity of the hygropause

- P21, L4. These results are fairly useful, and it would be good to expand a bit on the specifics, meaning what sort of change in resolution leads to a 50% change in amplitude? It is somewhat unfortunate also that you do not give a specific example or two whereby this aspect could really help to make the comparisons of amplitue (or phase) better between specific retrievals. - L21, delete "the data sets used here are based on" - L24, In a similar way, the observations are affected by aerosols.

- P22, L10, only on the latitude range - L14, polewards of 45 - L15, add a comma after variation - L17, accompanied by a substantial decrease of the agreement - L23, MIPAS data sets increased significantly polewards of 35... - L29, There are many ways to derive the characteristics of different variability patterns. - L32, add a comma after "Sect. 5.2".

- P23, L18, shows a sample comparison - L33, which show both negative ... The differences between ...

- P24, L25, add a comma after altitude. Also, add "and" before "in extreme cases". - L30/31, I would delete "as the variability in those is typically large" [or clarify]. - L33, change "could be" to "are".

- P25, L4, Other reasons include the different time periods used... [and delete "also contribute to the differences" at end of sentence]. - L12, tape recorder, that still exhibits fundamental differences versus observations,...

- Fig. 5 caption, Line 4, change "with respect of the MLS" to "with respect to the MLS".

- Fig. 6 caption, sentence 3, change "were" to "where".

- Fig. 9 caption, change Fig to Figs

- Supplement text, page 1, 3 lines from bottom, also exhibit some outliers (?). Also, Fig. 2 caption there, change "were" to "where", and in Fig. 3, change "phased" to "phase".
* * *

---

## Author Comment (AC1) · 17 Feb 2017

Comment #1: WAVAS II is an updated evaluation of the quality of available water vapor data sets for the middle atmosphere. I contributed the historic Nimbus 7 LIMS Version 5 data set to the first SPARC WAVAS report of 2000. After that time, a LIMS Version 6 data set was created and evaluated and made available for use by the research community, and this newer version has some subtle, but important improvements for the overall distributions of its stratospheric water vapor. Those improvements are reported in Remsberg et al. (2009), in addition to the initial findings from LIMS version 5 that you cite (Remsberg et al., 1984). Although you do not consider the LIMS data set for your current WAVAS II assessment, I respectfully request that you include the citation to Remsberg et al. (2009) on page 3, line 26 of your WAVAS II manuscript in place of Remsberg et al. (1984).

Remsberg, E. E., M. Natarajan, G. S. Lingenfelser, R. E. Thompson, B. T. Marshall, and L. L. Gordley, On the quality of the Nimbus 7 LIMS version 6 water vapor profiles and distributions, Atmos. Chem. Phys., 9, 9155-9167, doi:10.5194/acp-9-9155-2009, 2009.

Response #1:

Dear Ellis!

The citation of Remsberg et al. (1984) on page 3 was meant in a historical context, not in terms of data quality. To our knowledge this is the first publication showing aspects of the seasonal variation in the entire stratosphere. This refers to Fig.1 and 4 showing the latitude-altitude distribution for May and December. Thus we are hesitant to replace the initial reference. As a compromise we now refer both to Remsberg et al. (1984) and Remsberg et al. (2009) in the appropriate order, acknowledging the new information on the seasonal variation provided in the latter publication.

---

## Author Comment (AC2) · 17 Feb 2017

Replies to the Comments:

The authors thank the reviewers for their insightful comments. In the following, the comments are included in black while our replies are given in blue.

General comments:

The paper presents a very careful and detailed description of the annual, semiannual and quasi-biennial variations in middle-atmosphere water vapour, as observed by a large suite of satellite instruments. It is a very useful summary and should be published subject to some minor and technical corrections.

The written English is mostly very good. It has signs of being written by a non-native speaker (which is fine) but also has a few sentences which make no sense; these should be fixed. The writer has a tendency towards over-long sentences with inadequate punctuation; it would benefit from an attempt to improve the wording in these cases.

The figures are produced to a good standard and are very much improved from the pre-discussion technical review. I note that the zonal mean plots (e.g. Figure 1) appear in a wide variety of aspect ratios. In Figure 1 the plots are taller than they are wide, in Figure 5 (and much of the appendix) they are wider than they are tall, and in Figure 15 they are almost square. This makes it harder than it needs to be for the reader to compare one figure with another.

General response #1: In general this is owed to the number of panels in a figure. Figure 1 - 5, 6 - 9 and 11 - 13 use an A5 landscape format. We expect that these will be spread over the two columns in the journal version. There are differences in the width depending on the need for a y-axis label. Height-wise there a differences based on the colour bar and the legend (with a varying number of data sets). Figure 5, 10, 14 use an A5 portrait format and should only cover one column in the journal version. Figure 15 and those in the supplement use an A4 portrait. So far all figures used a 12cm width in the "\includegraphics" command, expect Fig 14 (which was wider unintentionally) and those in the supplement.

The typesetting of equations is not very good, especially considering that it was done with LaTeX. In particular:

• You do not need a dot for "multiply" and especially not between 2, π and t.

General response #2: We are aware that this is not necessary. It is simply a personal preference to be more explicit.

• Function names sin, cos, tan, atan, etc. should be set in upright font. Use \sin, \cos etc. in LaTeX and \mathrm{fnord} for a function fnord which is too obscure for LaTeX to know about.

General response #3: No problem! We have adapted that.

- It is not made clear why a number of quantities are typeset in bold.

General response #4: It was intended to indicate an altitude and/or latitude dependence. This information is now given after the first equation in Sect. 3.1. In principle also the time $t$ has such a dependence as the individual data sets lack coverage here and there. For simplicity we decided not to indicate that.

- It is not good style to use long words like "offset" and "linear" as subscripts. If you insist on doing it, they too should be in upright font: C_\mathrm{offset} to give Coffset.

General response #5: As before this is a personal preference. The more obvious the better. All subscripts are now upright, with the exception of the indices.

Specific comments:

Comment #1: Page 2, Lines 3–4: This sentence either makes no sense or the word order is very strange.

Response #1: The sentence has been rewritten as follows: "In general differences in the temporal variation of systematic errors and in the observational sampling play a central role."

Comment #2: Page 2 Line 12: The AMT house style is to write 0.3 W m−2, not 0.3 W/m2.

Response #2: Changed.

Comment #3: Page 3, Line 2: Between "dehydration" and "the" strikes me as a good place to use a colon (:).

Response #3: Okay, why not? Changed.

Comment #4: Page 3, Line 21: Space should be after comma, not before it.

Response #4: Thanks! It has been corrected.

Comment #5: Page 3, Lines 23–24: This sentence is particularly oddly worded and should be re-written.

Response #5: The sentence has been rewritten as follows: "A complete understanding of water vapour changes requires also a good knowledge of short term variability, such as the annual and semi-annual variation or the variation caused by the quasi-biennial oscillation (which we denote here as QBO variation)."

Comment #6: Page 5, Line 18. The use of "Be reminded" seems odd. Maybe replace with "Note"?

Response #6: It was actually more meant as a remainder than a note. There has been a discussion before it this sentence is really necessary, if it is not obvious. At the time we decided to keep it to be on the safe side. We retain this decision now.

Comment #7: Page 8, Line 4 and Page 15, Line 21: In both cases, "likely" should be replaced with "probably". Despite ending in "ly", the word "likely" is an adjective (synonymous with "probable") and should not be used as an adverb.

Response #7: Thanks for the advice. The word has been replaced.

Comment #8: Page 9 Lines 1–2: Box 2 in Figure 1 is in the middle and upper stratosphere, not the middle and lower stratosphere.

Response #8: Corrected.

Comment #9: Page 11, Line 21: "reveals" should read "is revealed".

Response #9: Corrected.

---

## Author Comment (AC3) · 17 Feb 2017

The comment was uploaded in the form of a supplement:
http://www.atmos-meas-tech-discuss.net/amt-2016-347/amt-2016-347-AC3-supplement.pdf